# Progressive Binarization with Semi-Structured Pruning for LLMs

## Abstract

Large language models (LLMs) have achieved remarkable progress in natural language processing, but their high computational and memory costs hinder deployment on resource-constrained devices. Binarization represents the most extreme form of quantization, yet binarized models still contain redundancy that can be further removed. Pruning provides a natural way to eliminate such redundancy, but naïve combination with binarization often results in severe performance degradation. In this paper, we propose Progressive Binarization with Semi-Structured Pruning (PBS$^2$P), a novel post-training framework that seamlessly integrates binarization and semi-structured pruning. We first propose Stepwise semi-structured Pruning with Binarization Optimization (SPBO), which progressively introduces sparsity while optimizing binarization parameters to jointly reduce pruning and quantization error, yielding more stable and accurate compression. Additionally, we propose a Coarse-to-Fine Search (CFS) that first allocates pruning ratios and then refines element selection, further enhancing overall performance. Extensive experiments across multiple LLM families show that PBS$^2$P consistently outperforms state-of-the-art (SOTA) binary post-training quantization methods in both perplexity and downstream accuracy. We will release all the code and models.

## 1 Introduction

Transformer-based large language models (LLMs) (Vaswani, 2017) have demonstrated remarkable performance across a wide range of natural language processing (NLP) tasks. This success is largely driven by their massive scale, often with billions of parameters. For instance, the OPT series (Zhang et al., 2022) includes models with up to 66 billion parameters, while the LLaMA family (Touvron et al., 2023a) features even larger models, such as LLaMA3-70B (Dubey et al., 2024). Although these models have substantially advanced the state of the art (SOTA) in NLP, their growing computational and memory demands present increasingly significant challenges for real-world deployment.

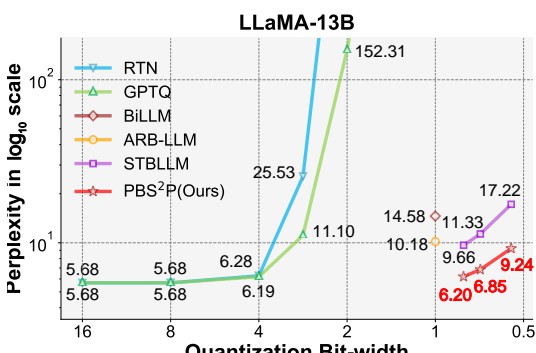

Figure 1: Perplexity of LLaMA-13B on WikiText2 under different bit-widths across various methods.

The compression of LLMs can be broadly categorized into several approaches, including weight quantization (Lin et al., 2024; Frantar et al., 2023), low-rank factorization (Zhang et al., 2024; Yuan et al., 2023), network pruning (Sun et al., 2024; Frantar & Alistarh, 2023), and knowledge distillation (Zhong et al., 2024; Gu et al., 2024). Among these, binarization represents an extreme form of quantization, reducing model weights to 1 bit and drastically lowering memory consumption. Recent efforts primarily focus on post-training quantization (PTQ) (Huang et al., 2024; Li et al., 2025; Dong et al., 2025), which eliminates the need for backpropagation, thereby accelerating the binarization process and improving deployment efficiency. For example, BiLLM (Huang et al., 2024) introduces a residual approximation strategy to enhance the performance of 1-bit LLMs, while ARB-LLM (Li et al., 2025) adopts an alternating refinement approach to better align binarized weights with their full-precision counterparts. STBLLM (Dong et al., 2025) further advances this

line of work by compressing LLMs to sub-1-bit precision. Despite the aforementioned advancements, binarized LLMs continue to exhibit redundancy, indicating opportunities for additional compression.

Pruning (LeCun et al., 1989) is a promising technique for further reducing redundancy in binarized models. However, conventional structured pruning (Ma et al., 2023; Ashkboos et al., 2024; Xia et al., 2024; An et al., 2024) often leads to severe performance degradation in LLMs. On the other hand, unstructured pruning (Dong et al., 2024) suffers from poor hardware compatibility and inefficient storage. Semi-structured pruning (Frantar & Alistarh, 2023; Sun et al., 2024; Dong et al., 2025) has emerged as an effective compromise, significantly reducing redundancy while maintaining a balance between accuracy and hardware efficiency. Nonetheless, directly combining binarization with semi-structured pruning often leads to noticeable performance degradation, as the joint constraints on weight values and structure increase the difficulty of preserving model accuracy under extreme compression. This problem is further complicated by the need to select pruning elements carefully, since suboptimal choices may remove important weights and amplify quantization errors. Therefore, designing a unified framework that balances compression and accuracy retention remains challenging.

To address these challenges, we propose **P**rogressive **B**inarization with **S**emi-**S**tructured **P**runing (PBS$^2$P) for LLMs, which achieves substantial model compression while maintaining strong performance (see Figure 1). We first propose S̲tepwise semi-structured P̲runing with B̲inarization O̲ptimization (SPBO), which progressively prunes a subset of elements at each step while jointly optimizing the binarized parameters. This strategy effectively reduces the cumulative error from pruning and binarization. To further enhance pruning efficiency, we develop a C̲oarse-to-F̲ine S̲earch (CFS) algorithm for more accurate pruning element selection. In the coarse stage, pruning ratios are assigned to each layer based on its importance. In the fine stage, a Hessian-based metric is used to identify the most redundant elements, guided by the layer-specific pruning ratios. We further show that the Hessian-based metric minimizes the theoretical error increase under a second-order approximation. This makes it a provably optimal strategy for pruning. Together, SPBO and CFS form the core of PBS$^2$P, enabling provably efficient compression with minimal accuracy degradation.

Extensive experiments demonstrate that PBS$^2$P consistently achieves SOTA performance across multiple LLM families, clearly surpassing existing binary PTQ methods on a wide range of evaluation benchmarks. As shown in Figure 1, on the WikiText-2 (Merity et al., 2017) dataset, PBS$^2$P attains a perplexity of 6.20 on LLaMA-13B (Touvron et al., 2023a) with an average bit-width of just 0.8 bits, compared to 5.47 for the full-precision model. These results further highlight the effectiveness of PBS$^2$P in significantly narrowing the performance gap between binarized and full-precision models.

Our key contributions can be summarized as follows:

- We propose a novel framework, PBS$^2$P, which integrates binarization and semi-structured pruning seamlessly for effective LLM compression.

- We propose S̲tepwise semi-structured P̲runing with B̲inarization O̲ptimization (SPBO), which progressively prunes the model while jointly optimizing binarization parameters, effectively reducing the combined error and preserving performance.

- We propose C̲oarse-to-F̲ine S̲earch (CFS) for selecting pruning elements, which enables effective identification of redundant parameters while better preserving model performance.

- Extensive experiments show that PBS$^2$P consistently outperforms SOTA binary PTQ methods and significantly narrows the gap to full-precision models across diverse benchmarks.

## 2 RELATED WORKS

### 2.1 LLM QUANTIZATION

Quantization compresses full-precision parameters into lower-bit representations, reducing both computation and storage demands. Current quantization methods for LLMs are mainly divided into Quantization-Aware Training (QAT) and Post-Training Quantization (PTQ). QAT (Liu et al., 2024; Chen et al., 2024; Du et al., 2024) integrates quantization during the training phase to enhance low-bit weight representations. However, due to the enormous parameter number, retraining becomes excessively expensive and inefficient for LLMs. PTQ, as it directly applies quantization to the model weights without retraining, making it faster and less resource-demanding. Recent methods, like ZeroQuant Yao et al. (2022) and BRECQ (Li et al., 2021), improve quantization accuracy by incorporating custom quantization blocks and group labels. While GPTQ (Frantar et al., 2023) and QuIP (Chee et al., 2024) use second-order error compensation to reduce quantization errors.

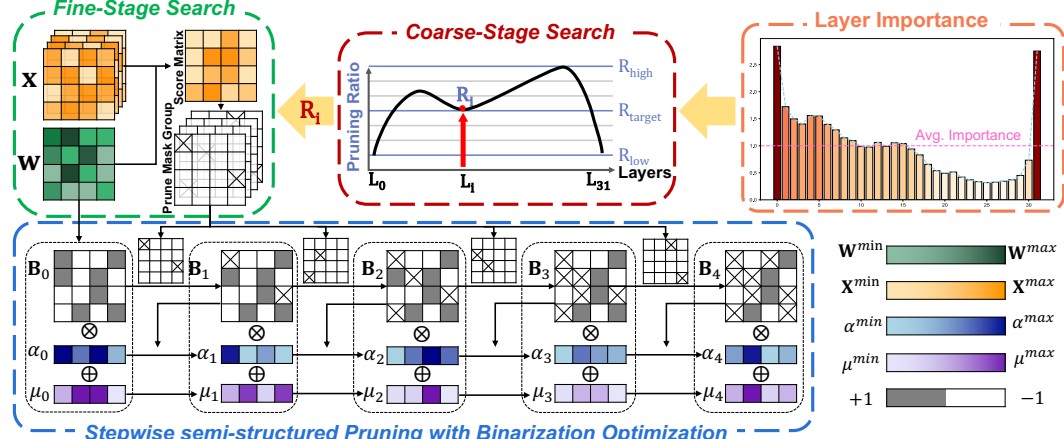

Figure 2: Overview of our PBS²P framework. **Coarse-Stage Search**: using the layer importance to assign pruning ratios to each layer. **Fine-Stage Search**: searching the elements to be pruned based on the Hessian-based metrics. **Stepwise semi-structured Pruning with Binarization Optimization**: stepwise pruning with optimization of binarized parameters.

## 2.2 NETWORK BINARIZATION

Binarization, as the most extreme form of quantization, reduces model parameters to a single bit (±1). Prominent methods, like Binary Weight Network (BWN) (Rastegari et al., 2016), XNOR-Net (Rastegari et al., 2016) and Bi-Real Net(Liu et al., 2018), focus on binarizing the weights, with XNOR-Net (Rastegari et al., 2016) and Bi-Real Net(Liu et al., 2018) also binarizing activations. In the context of LLM binarization, BitNet (Wang et al., 2023), OneBit (Xu et al., 2024), and BinaryMoS (Jo et al., 2024) adopt the QAT framework, while BiLLM (Huang et al., 2024), ARB-LLM (Li et al., 2025), and STBLLM(Dong et al., 2025) use PTQ combined with residual approximation. Focusing on the PTQ setting, our method yields significant enhancements over prior SOTA binary PTQ techniques.

## 2.3 MODEL PRUNING

Pruning (LeCun et al., 1989) is a common technique for compressing neural networks by removing less important parameters, resulting in smaller and more efficient sparse models. In LLMs, pruning methods are typically categorized as structured, unstructured, or semi-structured. Structured pruning (Ma et al., 2023; Ashkboos et al., 2024; Xia et al., 2024; An et al., 2024) removes entire components to enhance efficiency but often causes performance drops, requiring retraining. Unstructured pruning (Dong et al., 2024) eliminates individual weights based on importance, preserving performance even at high sparsity, though the irregular patterns are hardware-unfriendly. Semi-structured pruning offers a good trade-off, enforcing regular patterns like $N{:}M$ sparsity for hardware efficiency, as used in SparseGPT (Frantar & Alistarh, 2023), Wanda (Sun et al., 2024), and STBLLM (Dong et al., 2025). We adopt $N{:}M$ sparsity to balance performance and hardware efficiency.

## 3 METHODOLOGY

### 3.1 PRELIMINARY

**Binarization.** We begin by revisiting the standard binarization process (Rastegari et al., 2016). The full-precision matrix $\mathbf{W} \in \mathbb{R}^{n \times m}$ is first row-wise normalized by subtracting the mean of each row, yielding a zero-centered matrix $\widetilde{\mathbf{W}} \in \mathbb{R}^{n \times m}$:

$$\widetilde{\mathbf{W}} = \mathbf{W} - \mu, \quad \text{where} \quad \mu = \frac{1}{m} \sum_{j=1}^{m} \mathbf{W}_{\cdot j}. \tag{1}$$

This preprocessing step mitigates distributional bias and promotes a more symmetric weight distribution, which is beneficial for subsequent binarization. The goal of binarization is to minimize the quantization error: $\arg\min_{\alpha,\mathbf{B}} \|\widetilde{\mathbf{W}} - \alpha\mathbf{B}\|_F^2$, where $\alpha \in \mathbb{R}^n$ is a row-wise scaling vector and $\mathbf{B} \in \{+1, -1\}^{n \times m}$ is the binary matrix. The closed-form solutions are given by $\alpha = \frac{1}{m} \sum_{j=1}^{m} |\widetilde{\mathbf{W}}_{\cdot j}|$, which computes the mean absolute value across each row, and $\mathbf{B} = \text{sign}(\widetilde{\mathbf{W}})$, which binarizes the weights via sign function (Huang et al., 2024).

**N:M Sparsity.**  $N{:}M$ sparsity is a semi-structured pruning scheme that balances model compression and hardware efficiency. Unlike unstructured pruning, which removes arbitrary weights and results in irregular sparsity patterns that are hard to accelerate, $N{:}M$ sparsity imposes a structured constraint by retaining exactly $N$ nonzero elements out of every $M$ consecutive weights (Frantar & Alistarh, 2023; Sun et al., 2024; Dong et al., 2025). A typical configuration, such as 2:4 sparsity, keeps only two weights out of every four, significantly reducing parameter count while preserving computational efficiency. This regular pattern allows for efficient execution on NVIDIA Ampere GPUs (Nvidia, 2020), taking advantage of specialized hardware support. In our work, we adopt the $N{:}M$ sparsity scheme as a semi-structured pruning strategy to further eliminate redundancy within the binarized model. This enables stronger compression with hardware-friendly execution.

## 3.2 Stepwise Semi-Structured Pruning with Binarization Optimization

To facilitate the joint optimization of binarization and semi-structured pruning, we formulate the total compression error $\mathcal{L}$ as the objective function to be minimized:

$$\mathcal{L} = ||\mathbf{W}\mathbf{X} - (\mathbf{M}_\mathrm{p} \odot \widehat{\mathbf{W}})\mathbf{X}||_F^2, \quad \text{where } \widehat{\mathbf{W}} = \alpha\mathbf{B} + \mu\,. \tag{2}$$

where $\mathbf{X}$ denotes the calibration data, $\mathbf{M}_\mathrm{p}$ denotes the semi-structured pruning mask. Optimizing both the pruning mask $\mathbf{M}_\mathrm{p}$ and the binarized parameters $\widehat{\mathbf{W}}$ simultaneously is an NP-hard problem (Blumensath & Davies, 2008). Therefore, we adopt a greedy-style approximation. At each step, we prune a small portion of elements. Then, we update the binarized parameters to fit the current pruning ratio. This step-by-step strategy reduces the optimization difficulty. It distributes optimization across multiple pruning stages. Selection of the semi-structured mask $\mathbf{M}_\mathrm{p}$ is detailed in Section 3.3.

**N: M Sparsity Pruning Mask Group.** To support progressive pruning, we decompose the original pruning mask $\mathbf{M}_\mathrm{p}$ into a sequence of sub-masks, each corresponding to one intermediate pruning step. Under the $N{:}M$ sparsity constraint, exactly $N$ out of every $M$ elements are retained. Within each $M$-element group, the $M{-}N$ pruned elements are removed sequentially across successive stages. Formally, the

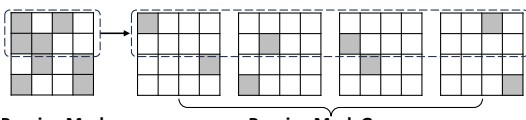

**Pruning Mask**          **Pruning Mask Group**

Figure 3: Illustration of the $N{:}M$ sparsity pruning mask group, where the original pruning mask $\mathbf{M}_\mathrm{p}$ is divided into $M{-}N$ sub masks.

full pruning mask can be expressed as the union of $M{-}N$ sub-masks: $\mathbf{M}_\mathrm{p} = \bigcup_{i=1}^{M-N} \mathbf{M}_\mathrm{p}^i$. This decomposition enables fine-grained progressive pruning and allows for a tighter and more seamless integration with binarization in subsequent stages, as illustrated in Figure 3.

**Stepwise Pruning with Binarization Optimization (SPBO).**  After obtaining the group of pruning masks, we proceed with the stepwise pruning and binarization optimization process. We perform a total of $M{-}N$ pruning steps to achieve $N{:}M$ sparsity. After each step, we update the binarization parameters $\alpha$ and $\mu$ to better adapt to the current pruning state. At the $k$-th pruning step, we obtain the updated binary matrix $\mathbf{B}_k$ and full-precision matrix $\mathbf{W}_k$ by applying the $k$-th pruning sub-mask $\mathbf{M}_\mathrm{p}^k$ to the previously pruned matrices $\mathbf{B}_{k-1}$ and $\mathbf{W}_{k-1}$:

$$\mathbf{B}_k = \mathbf{B}_{k-1} \odot \mathbf{M}_\mathrm{p}^k, \quad \mathbf{W}_k = \mathbf{W}_{k-1} \odot \mathbf{M}_\mathrm{p}^k. \tag{3}$$

After the $k$-th pruning step, the binarization parameters $\alpha$ and $\mu$ are no longer optimal under the updated sparsity pattern. To minimize the error $\mathcal{L}$ in the current pruning state, these parameters must be re-estimated. Inspired by Li et al. (2025), a straightforward approach is to compute the partial derivatives of $\mathcal{L}$ with respect to $\alpha$ and $\mu$, and solve for the stationary points by setting them to zero. Specifically, we first update $\mu$ by setting $\partial\mathcal{L}/\partial\mu = 0$ during the $k$-th pruning step:

$$\partial\mathcal{L}/\partial\mu = 0 \quad \Rightarrow \quad \mu = \frac{\mathbf{1}^\top \mathbf{S}(\mathbf{W}_k - \alpha\mathbf{B}_k)^\top}{\mathbf{1}^\top \mathbf{S}\mathbf{1}}, \tag{4}$$

where $\mathbf{S} = \sum_b \mathbf{X}_b^\top \mathbf{X}_b$ is a precomputed matrix derived from the calibration data $\mathbf{X}$ for notational convenience. Then, we adjust $\alpha$ by setting $\partial\mathcal{L}/\partial\alpha = 0$:

$$\partial\mathcal{L}/\partial\alpha = 0 \quad \Rightarrow \quad \alpha = \frac{\mathrm{diag}(\mathbf{B}_k\mathbf{S}(\mathbf{W}_k - \mu)^\top)}{\mathrm{diag}(\mathbf{B}_k\mathbf{S}\mathbf{B}_k^\top)}. \tag{5}$$

During the $k$-th pruning round, we can alternately update $\mu$ and $\alpha$ for $\tau$ steps to further reduce $\mathcal{L}$. We perform total $M{-}N$ steps of pruning until all pruning elements are removed. The detailed derivation of optimizing $\mu$ and $\alpha$ can be found in the supplementary material.

### 3.3 COARSE-TO-FINE SEARCH

Selecting effective pruning elements is a key challenge in model pruning. To address this, we adopt a two-stage strategy. In the coarse stage, we assign pruning ratios to each layer based on its layer importance. In the fine stage, we determine the specific elements to prune within each layer according to the assigned ratios, allowing for more accurate selection of pruning candidates.

**Coarse-Stage: Pruning Ratio Allocation.** To estimate the importance of each layer, inspired by Dumitru et al. (2024), we use cosine similarity to measure the alignment between its input and output representations. We adopt a straightforward criterion that quantifies how strongly a layer shapes the input–output relationship. Layers that induce larger changes are regarded as more important, whereas those with minimal effect are considered less important. Specifically, for a given weight layer $\mathcal{W}$, let $\mathcal{X}_{\mathcal{W}}$ and $\mathcal{Y}_{\mathcal{W}}$ denote its input and output, respectively. The importance of the layer is then computed as follows:

$$\text{Importance}(\mathcal{W}) = \frac{1}{\text{similarity}(\mathcal{X}_{\mathcal{W}}, \mathcal{Y}_{\mathcal{W}}) + 1}, \text{ where similarity}(\mathcal{X}_{\mathcal{W}}, \mathcal{Y}_{\mathcal{W}}) = \frac{\mathcal{X}_{\mathcal{W}} \cdot \mathcal{Y}_{\mathcal{W}}}{\|\mathcal{X}_{\mathcal{W}}\|_2 \|\mathcal{Y}_{\mathcal{W}}\|_2}. \quad (6)$$

Based on the computed importance, we rank all layers, with $k_i$ denoting the rank of the $i$-th layer. The pruning parameter $N_i$ for each layer is determined as follows:

$$N_i = \left\lfloor N_{\text{high}} - (N_{\text{high}} - N_{\text{low}}) \cdot \frac{k_i - 1}{L - 1} + \frac{1}{2} \right\rfloor, \quad (7)$$

where $L$ represents the total number of layers in the model. The pruning bounds $N_{\text{high}}$ and $N_{\text{low}}$ are derived from the target pruning parameter $N_{\text{target}}$, using a simple and stable rule: $N_{\text{high}} = N_{\text{target}} + 1$ and $N_{\text{low}} = N_{\text{target}} - 1$. This design provides a narrow yet effective range for allocating different pruning strengths across layers. Consequently, the most important layer (e.g., the $m$-th layer with $k_m = 1$) is assigned $N_{\text{high}}$, retaining more elements, while the least important layer (e.g., the $n$-th layer with $k_n = L$) is assigned $N_{\text{low}}$, preserving fewer elements.

**Fine-Stage: Selecting Pruning Elements.** Once the pruning ratio $N/M$ is assigned in the coarse-stage, we proceed to search for the specific pruning elements within each layer. We revisit the impact of pruning on model error. Inspired by Hassibi et al. (1993), we derive the relationship between pruning and error increase under a second-order approximation, and present the corresponding formulation in Theorem 3.1. The full proof is provided in the supplementary material. Since our objective is to minimize the increase in error caused by pruning, we directly adopt the following score to guide element selection: $s_q = \frac{w_q^2}{[\mathbf{H}^{-1}]_{qq}^2}$. This criterion is theoretically guaranteed to minimize the error increase introduced by pruning under a second-order approximation. Based on the given $N{:}M$ ratio and the corresponding $s_q$ values, we select the top $N$ elements with the largest $s_i$ values from every $M$ consecutive elements to retain. The other $M - N$ elements are selected for pruning.

> **Theorem 3.1.** *After pruning an element $w_q$, the pruning-induced error increase $\delta\mathcal{L}$ can be approximated by the following expression:*
>
> $$\delta\mathcal{L} = \frac{1}{2} \frac{w_q^2}{[\mathbf{H}^{-1}]_{qq}}, \quad (8)$$
>
> *where $[\mathbf{H}^{-1}]_{qq}$ denotes the $q$-th diagonal entry of the inverse Hessian matrix.*

### 3.4 PBS²P PIPELINE

**PBS²P Workflow.** As illustrated in Figure 2, the proposed PBS²P framework is composed of three key components: Coarse-stage Search, Fine-stage Search, and Stepwise Semi-structured Pruning with Binarization Optimization. The workflow of PBS²P is outlined in Algorithm 1. Given a full-precision model $\mathcal{M}$, calibration data $\mathcal{X}$, and a target average pruning ratio $\mathcal{TR}$, the algorithm first performs a coarse-stage search to determine layer-wise pruning ratios based on importance. With the ratios determined, we proceed layer by layer to apply pruning and binarization. For each layer $l$, we extract its weight matrix $\mathcal{W}_l$ and conduct a fine-stage search to generate a semi-structured pruning mask $\mathbf{M}_{\text{p}}$. The SPBO procedure is then invoked to progressively prune the weights while optimizing the binarization parameters. This joint optimization reduces the total error introduced by pruning and binarization. The updated weights $\mathcal{W}_l'$ are written back to the model, and the process repeats for all layers. As pruning and binarization are applied layer-wise, our method has low memory overhead.

---

**Algorithm 1** Pseudocode of PBS$^2$P

---

1: **Inputs:** Model $\mathcal{M}$, Calibration Data $\mathcal{X}$, Target Pruning Ratio $\mathcal{TR}$, Optimization Steps $\tau$
2: **Output:** Binarized and Pruned Model $\mathcal{M}'$
3: **procedure** PBS$^2$P$(\mathcal{M}, \mathcal{X}, \mathcal{TR}, \tau)$
4:     $R \leftarrow$ COARSE_STAGE_SEARCH$(\mathcal{M}, \mathcal{X}, \mathcal{TR})$     ▷ Allocate layer-wise pruning ratios
5:     **for** each layer $l$ in model $\mathcal{M}$ **do**
6:         $\mathcal{W}_l \leftarrow$ EXTRACT_WEIGHTS$(\mathcal{M}, l)$     ▷ Obtain weights of layer $l$
7:         $r_l \leftarrow R[l]$     ▷ Get pruning ratio for layer $l$
8:         $\mathbf{M}_{\mathrm{p}} \leftarrow$ FINE_STAGE_SEARCH$(\mathcal{W}_l, \mathcal{X}, r_l)$     ▷ Generate pruning mask
9:         $\mathcal{W}_l' \leftarrow$ SPBO$(\mathcal{W}_l, \mathcal{X}, \mathbf{M}_{\mathrm{p}}, \tau)$     ▷ Stepwise pruning and binarization optimization
10:         $\mathcal{M}' \leftarrow$ UPDATE_LAYER$(\mathcal{M}, l, \mathcal{W}_l')$     ▷ Write updated weights back to model
11:     **end for**
12:     **return** $\mathcal{M}'$
13: **end procedure**

---

**Average Bits.** Following BiLLM (Huang et al., 2024) and STBLLM (Dong et al., 2025), we similarly divide weights into salient and non-salient groups, consistent with their design. For salient weights, we adopt a residual binarization strategy, allocating two bits to better preserve their expressiveness, as done in both prior works. For non-salient weights, we apply group-wise quantization. In line with STBLLM (Dong et al., 2025), we further divide the non-salient weights into three groups using two split points, maintaining the same quantization scheme. The weight parameters and additional hardware overhead are as follows:

$$\begin{cases} N_{\mathrm{param}} = [2 \times r_{\mathrm{salient}} + (1 - r_{\mathrm{salient}})] \times \frac{N}{M}, \\ N_{\mathrm{storing}} = 2 + \dfrac{1}{b_{\mathrm{size}}}, \end{cases} \tag{9}$$

where $r_{\mathrm{salient}}$ represents the proportion of salient weights, $\frac{N}{M}$ denotes the predefined average pruning ratio for the entire model, and $b_{\mathrm{size}}$ indicates the block size used in OBC (Frantar & Alistarh, 2022) compensation, with 2 bits reserved to mark the division between salient and non-salient weights. Our parameter settings and $\frac{N}{M}$ configuration are identical to our main comparison method STBLLM (Dong et al., 2025). Specifically, we adopt the same settings of 4:8, 5:8, and 6:8 sparsity as used in STBLLM (Dong et al., 2025). These correspond to average bit-widths of 0.55, 0.70, and 0.80, respectively. This alignment ensures fair, direct comparison under consistent compression budgets.

## 4 EXPERIMENTS

### 4.1 SETTINGS

All experiments are performed using PyTorch (Paszke et al., 2019b) and Huggingface (Paszke et al., 2019a) on a single NVIDIA A800-80GB GPU. Following the work of (Frantar et al., 2023), (Huang et al., 2024), and (Li et al., 2025), we use 128 samples from the C4 (Raffel et al., 2020) dataset for calibration. Since PBS$^2$P is an efficient PTQ framework, it eliminates fine-tuning, enabling completion through a single process combining binarization and pruning.

**Models and Datasets.** We conduct extensive experiments on the LLaMA (Touvron et al., 2023a), LLaMA-2 (Touvron et al., 2023b), and LLaMA-3 (Dubey et al., 2024) families and the OPT family (Zhang et al., 2022). To evaluate the effectiveness of PBS$^2$P, we measure the perplexity of LLM outputs on WikiText2 (Merity et al., 2017), PTB (Marcus et al., 1994), and C4 (Raffel et al., 2020). Moreover, we also evaluate the accuracy on seven zero-shot QA datasets: ARC-c (Clark et al., 2018), ARC-e (Clark et al., 2018), BoolQ (Clark et al., 2019), Hellaswag (Zellers et al., 2019), OBQA (Mihaylov et al., 2018), RTE (Chakrabarty et al., 2021), and Winogrande (Sakaguchi et al., 2020). This benchmark setup validates our method on both language modeling and reasoning tasks.

**Baselines.** We mainly compare our PBS$^2$P with STBLLM (Dong et al., 2025), a structural binary PTQ framework designed for compressing LLMs to precisions lower than 1-bit. We compare the results of PBS$^2$P with STBLLM under the same $N{:}M$ sparsity settings (*e.g.*, 4:8, 5:8, 6:8). Previous low-bit methods like ARB-LLM (Li et al., 2025), BiLLM (Huang et al., 2024), GTPQ (Frantar et al., 2023), and vanilla RTN are also selected as baselines for comparison.

Table 1: Perplexity comparison of RTN, GPTQ (Frantar et al., 2023), BiLLM (Huang et al., 2024), ARB-LLM (Li et al., 2025), STBLLM (Dong et al., 2025), and PBS$^2$P on the LLaMA families. The evaluation results demonstrate the perplexity performance on the Wikitext2 (Merity et al., 2017) dataset across various model sizes.

| Settings | | | LLaMA-1 | | | | LLaMA-2 | | LLaMA-3 |
|---|---|---|---|---|---|---|---|---|---|
| Method | #Block | W-Bits | 7B | 13B | 30B | 65B | 7B | 13B | 8B |
| FP16 | - | 16 | 5.68 | 5.09 | 4.10 | 3.53 | 5.47 | 4.88 | 6.14 |
| RTN | - | 3 | 25.54 | 11.40 | 14.89 | 10.59 | 542.80 | 10.68 | 2194.98 |
| GPTQ | 128 | 3 | 8.63 | 5.67 | 4.87 | 4.17 | 6.44 | 5.46 | 18.68 |
| RTN | - | 2 | 106767.34 | 57409.93 | 26704.36 | 19832.87 | 17788.94 | 51145.61 | 1335816.13 |
| GPTQ | 128 | 2 | 129.20 | 20.46 | 15.29 | 8.66 | 52.22 | 23.63 | 1480.43 |
| RTN | - | 1 | 168388.00 | 1412020.25 | 14681.76 | 65253.24 | 157058.34 | 47902.32 | 1353698.38 |
| GPTQ | 128 | 1 | 164471.78 | 131505.41 | 10339.15 | 20986.16 | 59758.69 | 22926.54 | 1121260.50 |
| BiLLM | 128 | 1.11 | 49.79 | 14.58 | 9.90 | 8.37 | 32.31 | 21.35 | 55.80 |
| ARB-LLM | 128 | 1.11 | 14.03 | 10.18 | 7.75 | 6.56 | 16.44 | 11.85 | 27.42 |
| STBLLM | 128 | 0.80 | 15.03 | 9.66 | 7.56 | 6.43 | 13.06 | 11.67 | 33.44 |
| STBLLM | 128 | 0.70 | 19.48 | 11.33 | 9.19 | 7.91 | 18.74 | 13.26 | 49.12 |
| STBLLM | 128 | 0.55 | 31.72 | 17.22 | 13.43 | 11.07 | 27.93 | 20.57 | 253.76 |
| PBS$^2$P | 128 | 0.80 | 7.36 | 6.20 | 5.21 | 4.60 | 7.17 | 6.27 | 10.75 |
| PBS$^2$P | 128 | 0.70 | 8.09 | 6.85 | 5.78 | 5.09 | 8.00 | 6.89 | 12.29 |
| PBS$^2$P | 128 | 0.55 | 10.78 | 9.24 | 7.19 | 6.39 | 10.64 | 8.68 | 17.45 |

## 4.2 MAIN RESULTS

We perform a comprehensive comparison of different LLM families (like LLaMA (Touvron et al., 2023a; Dubey et al., 2024; Touvron et al., 2023b) and OPT (Zhang et al., 2022)) with various model sizes. To keep fairness, we follow STBLLM (Dong et al., 2025) to report the average bit-width of all methods, where our methods have the same bit-width as STBLLM.

**Results on LLaMA Family.** As shown in Table 1, the models using RTN and GPTQ methods find it hard to maintain model performance at 1-bit precision. BiLLM achieves a satisfactory perplexity of 1.11 bits but performs worse than ARB-LLM at the same bit-width. At sub-1-bit precision, PBS$^2$P surpasses STBLLM and significantly reduces perplexity at the same bit-width across model sizes from 7B to 65B. For instance, PBS$^2$P achieves a substantial improvement over STB-LLM on LLaMA-1-7B, with perplexity dropping from 31.72 to 10.78, a reduction of approximately 66.0%, in the extreme case of 4:8 structured binarization, where half of the parameters are pruned.

Furthermore, PBS$^2$P, with a precision of 0.8 bits, outperforms both RTN at 3 bits, GPTQ at 2 bits, BiLLM and ARB-LLM at 1.11 bits, and STBLLM at 0.8 bits in terms of perplexity across all model sizes. Those comparisons show that our PBS$^2$P achieves a better trade-off between bit precision and performance. It is worth noting that PBS$^2$P outperforms GPTQ at 3 bits on LLaMA 1-7B and LLaMA 3-8B. We extend perplexity evaluation to the PTB and C4 datasets. Figure 4 shows the performance of the LLaMA-7B, LLaMA-13B, and LLaMA-2-7B

Table 2: Perplexity on the OPT family using Wiki2.

| Method | Block | W-Bit | 1.3B | 2.7B | 30B |
|---|---|---|---|---|---|
| FP16 | - | 16 | 14.62 | 12.47 | 9.56 |
| RTN | - | 2 | 11272.65 | 9505.76 | 1165864.25 |
| GPTQ | 128 | 2 | 121.64 | 59.53 | 13.04 |
| RTN | - | 1 | 17165.72 | 36516.69 | 6485.99 |
| GPTQ | 128 | 1 | 8719.58 | 11700.13 | 14083.15 |
| BiLLM | 128 | 1.11 | 69.05 | 48.61 | 13.86 |
| ARB-LLM | 128 | 1.11 | 26.63 | 19.84 | 11.12 |
| STBLLM | 128 | 0.80 | 29.84 | 17.02 | 12.80 |
| STBLLM | 128 | 0.70 | 33.01 | 20.82 | 14.38 |
| STBLLM | 128 | 0.55 | 45.11 | 30.34 | 18.80 |
| PBS$^2$P | 128 | 0.80 | 23.72 | 18.32 | 10.50 |
| PBS$^2$P | 128 | 0.70 | 27.10 | 20.17 | 10.82 |
| PBS$^2$P | 128 | 0.55 | 44.53 | 27.42 | 11.87 |

models. PBS$^2$P continues to achieve a leading edge in performance while consistently operating at a relatively lower bit-width compared to other methods across diverse evaluation settings.

**Results on OPT Family.** We extend our experiments to the OPT family (1.3B to 30B) under sub-1-bit PTQ settings, similar to the setup for LLaMA family. As shown in Table 2, PBS$^2$P continues to outperform STBLLM across most of the models and $N{:}M$ structured binarization configurations. More results are provided in the supplementary material.

## 4.3 ZERO-SHOT RESULTS

To provide a more thorough evaluation of binary LLMs, we extend our experiments to 7 zero-shot datasets and test on models from the LLaMA family: LLaMA-1-13B, LLaMA-2-13B, and LLaMA-1-30B. Each model is evaluated across various compression methods, including full-precision, STBLLM (6:8), STBLLM (4:8), PBS$^2$P (6:8), and PBS$^2$P (4:8). As shown in Table 3, models compressed with PBS$^2$P significantly outperform those compressed with STBLLM in terms of average accuracy. Such comparisons demonstrate that PBS$^2$P provides a more effective solution for compressing LLMs to

Table 3: Accuracies (%) for 7 zero-shot tasks from semi-structured binarized LLaMA-1-13B, LLaMA-2-13B, and LLaMA-1-30B with STBLLM and PBS$^2$P.

| Models | Method | W-Bits | Winogrande | OBQA | Hellaswag | Boolq | ARC-e | ARC-c | RTE | Average |
|---|---|---|---|---|---|---|---|---|---|---|
| | FP16 | 16 | 72.69 | 33.20 | 59.91 | 77.89 | 77.40 | 46.42 | 70.40 | 63.80 |
| | STBLLM | 0.80 | 65.98 | 36.20 | 63.67 | 65.38 | 68.86 | 34.04 | 56.68 | 55.83 |
| LLaMA-1-13B | STBLLM | 0.55 | 63.06 | 34.80 | 52.65 | 62.48 | 56.90 | 28.33 | 52.71 | 50.13 |
| | PBS$^2$P | 0.80 | 72.77 | 31.00 | 54.80 | 74.71 | 74.37 | 42.32 | 68.23 | 59.74 |
| | PBS$^2$P | 0.55 | 69.30 | 26.80 | 46.83 | 71.56 | 65.70 | 32.68 | 55.96 | 52.69 |
| | FP16 | 16 | 72.22 | 35.20 | 60.03 | 80.55 | 79.42 | 48.38 | 65.34 | 65.00 |
| | STBLLM | 0.80 | 63.93 | 37.00 | 57.76 | 71.53 | 60.56 | 31.99 | 54.15 | 53.85 |
| LLaMA-2-13B | STBLLM | 0.55 | 55.88 | 29.40 | 44.03 | 64.31 | 48.86 | 26.54 | 52.71 | 45.96 |
| | PBS$^2$P | 0.80 | 72.45 | 31.00 | 54.43 | 80.61 | 74.28 | 42.75 | 59.21 | 59.24 |
| | PBS$^2$P | 0.55 | 69.85 | 27.00 | 47.75 | 75.50 | 69.19 | 35.58 | 62.82 | 55.38 |
| | FP16 | 16 | 75.77 | 36.00 | 63.37 | 82.69 | 80.30 | 52.90 | 67.15 | 67.40 |
| | STBLLM | 0.80 | 71.59 | 41.00 | 69.85 | 77.37 | 71.55 | 41.30 | 48.01 | 60.10 |
| LLaMA-1-30B | STBLLM | 0.55 | 64.01 | 34.60 | 56.46 | 63.06 | 60.86 | 31.48 | 51.99 | 51.78 |
| | PBS$^2$P | 0.80 | 75.93 | 35.00 | 59.45 | 82.14 | 79.29 | 47.95 | 63.18 | 63.28 |
| | PBS$^2$P | 0.55 | 72.14 | 31.20 | 53.00 | 79.76 | 73.99 | 41.13 | 69.31 | 60.08 |

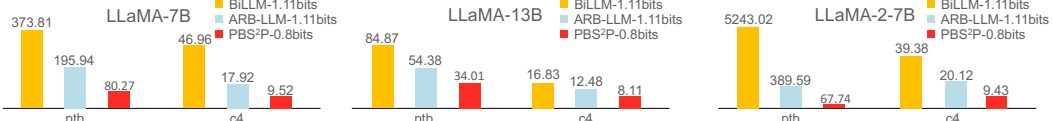

Figure 4: BiLLM (Huang et al., 2024), ARB-LLM (Li et al., 2025), and PBS$^2$P performed on the PTB (Marcus et al., 1994) and c4 (Raffel et al., 2020) datasets, mainly on LLaMA-7B, LLaMA-13B, and LLaMA2-7B, and we found that PBS$^2$P at 0.8 bits outperforms other methods.

less than 1-bit, offering improved trade-offs between accuracy and efficiency. This highlights its strong potential for enabling high-performance, low-bit inference in real-world applications.

### 4.4 ABLATION STUDY

**Ablation for SPBO Strategy.** To validate the effectiveness of our SPBO strategy, we provide the performance of PBS$^2$P with and without its application. As shown in Table 4a, the performance of SPBO surpasses that of the vanilla pruning followed by binarization approach. In the vanilla method, pruning is done in a single step, where all elements are pruned simultaneously, and binarization is applied to the remaining elements afterward. This approach makes it difficult to jointly minimize the combined pruning and binarization errors. In contrast, our SPBO strategy progressively couples these two processes, resulting in a clear performance improvement.

**Ablation for Metric in Fine-Stage Search.** The Table 4b presents the performance of different pruning metrics in the fine-stage search. We compare several metrics, including random selection, magnitude-based selection, Wanda, SI, and our Hessian-based metric. Our metric significantly outperforms both random selection and magnitude-based pruning, achieving superior performance compared to Wanda and SI, thereby highlighting the effectiveness of our approach.

**Ablation for Metric in Coarse-Stage Search.** Table 4c shows the performance of the coarse-stage search under three conditions: without the coarse-stage search, using the relative importance (RI) metric from STBLLM, and applying our layer importance (LI) score. The results reveal that using the RI metric causes a performance drop compared to the baseline without coarse-stage search. In contrast, our proposed layer importance (LI) score achieves substantial improvements, validating both the necessity of the coarse-stage search and the superiority of our metric.

**Ablation for Pruning Type.** In Table 4d, we present the impact of different pruning types on the results, comparing structured pruning, unstructured pruning, and semi-structured pruning under the same pruning ratio (50%). We adopt column pruning for the structured case, elementwise magnitude-based pruning for the unstructured case, and 4:8 sparsity for the semi-structured case. Semi-structured pruning strikes an optimal balance: it maintains hardware efficiency while achieving performance close to unstructured pruning and significantly better than structured pruning.

**Ablation for Group Size.** Table 4e presents the results of our ablation study on the group size configuration. It indicates that a smaller group size, meaning finer-grained grouping, leads to better performance. However, this also comes with increased computational and storage costs. To strike a balance between performance and resource efficiency, we select a group size of 128.

Table 4: Ablation study on LLaMA-7B, where all PBS$^2$P is applied an $N{:}M$ sparsity of 4:8. Results are measured by perplexity on the Wikitext2 and C4 datasets. Our results are highlighted in **bold**.

(a) Ablation for SPBO Strategy.

| SPBO Strategy | Wikitext2↓ | C4↓ |
|---|---|---|
| ✗ | 14.43 | 16.76 |
| ✓ | **10.78** | **13.06** |

(b) Ablation for Metric in Fine-Stage Search

| Metric | Random | Magnitude | Wanda | SI | Ours |
|---|---|---|---|---|---|
| Wikitext2↓ | 7,779.28 | 38.90 | 10.89 | 196.61 | **10.78** |
| C4↓ | 6,797.09 | 24.47 | 13.16 | 148.85 | **13.06** |

(c) Ablation for Metric in Coarse-Stage Search

| Coarse-Stage Search | Metric | Wikitext2↓ | C4↓ |
|---|---|---|---|
| ✗ | — | 11.55 | 13.94 |
| ✓ | RI | 14.43 | 16.76 |
| ✓ | LI | **10.78** | **13.06** |

(d) Ablation for Pruning Type

| Prune Type | Hardware-friendly | Wikitext2↓ | C4↓ |
|---|---|---|---|
| Structured | ✓ | 621.16 | 400.50 |
| Unstructured | ✗ | 8.54 | 10.95 |
| Semi-Structured | ✓ | **10.78** | **13.06** |

(e) Ablation for Group Size

| Group Size | 32 | 64 | 128 | 256 | 512 |
|---|---|---|---|---|---|
| Wikitext2↓ | 9.74 | 10.24 | **10.78** | 11.20 | 11.91 |
| C4↓ | 11.75 | 12.50 | **13.06** | 13.68 | 14.51 |

(f) Ablation for Calibration Data Size

| Calibration Data Size | 32 | 64 | 128 | 256 |
|---|---|---|---|---|
| Wikitext2↓ | 11.62 | 11.20 | **10.78** | 10.77 |
| C4↓ | 13.58 | 13.44 | **13.06** | 13.12 |

**Ablation for Calibration Data Size.** Similar to other PTQ-based methods, our PBS$^2$P framework requires only a small calibration set. Table 4f shows the effect of varying calibration size. With the C4 dataset, reducing the sample size to 32 or 64 degrades performance, while increasing to 256 yields minimal additional benefit over 128. PBS$^2$P remains robust with only 128 calibration samples.

### 4.5 TIME ANALYSIS

As a binary PTQ framework, PBS$^2$P eliminates the need for fine-tuning and retraining. It introduces coarse-to-fine search and stepwise pruning with binarization optimization, which moderately increases the overall computation time. As shown in Table 5, BiLLM (Huang et al., 2024) completes quantization on LLaMA-7B in 45 minutes, while ARB-LLM (Li et al., 2025) takes 76 minutes. PPBS$^2$P finishes in 111 minutes, with only 2 minutes spent on the coarse-stage search and 109 minutes on the fine-stage search with SPBO. This additional overhead is mainly due to iterative parameter updates in the fine stage. This overhead is acceptable since all computations are carried out offline, imposing no additional latency during inference. With this design, PBS$^2$P achieves superior accuracy and highly efficient, practical deployment.

Table 5: Time comparison of different PTQ methods on LLaMA-7B.

| Method | Coarse stage | SPBO | Total |
|---|---|---|---|
| BiLLM | – | – | 45 min |
| ARB-LLM | – | – | 76 min |
| PBS$^2$P | 2 min | 109 min | 111 min |

### 4.6 INFERENCE EFFICIENCY

We evaluate inference efficiency using custom matrix multiplication kernels for binarized and semi-structured sparse weights, built on top of the open-source BitBLAS library[1]. As shown in the figure 5, PBS²P (4:8) consistently achieves the lowest latency across typical LLM projection shapes, with activation sequence length fixed at 2048. Compared to FP16 and other 1-bit baselines like BiLLM and ARB-LLM, PBS²P delivers significant speedups, especially on wider matrices such as 4096×11008 and 11008×4096.

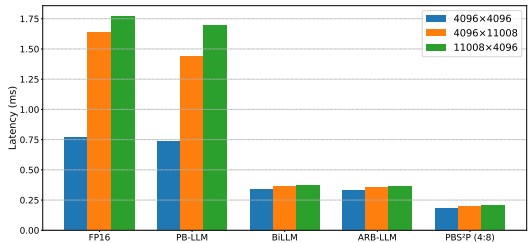

Figure 5: Latency comparison (ms) of matrix multiplications across different projection shapes.

### 5 CONCLUSION

In this work, we propose Progressive Binarization with Semi-Structured Pruning (PBS$^2$P) for LLM compression. Central to our approach is SPBO, a stepwise semi-structured pruning strategy with binarization optimization, which prunes a subset of elements at each step while jointly optimizing the binarized parameters. SPBO effectively mitigates the combined error from pruning and binarization. Additionally, we introduce a Coarse-to-Fine Search (CFS) strategy to enhance pruning element selection, further improving compression efficiency. Extensive experiments demonstrate that PBS$^2$P outperforms SOTA binary PTQ methods, delivering superior accuracy across various LLM families and evaluation metrics. Our method enables efficient LLM deployment on resource-limited devices while preserving performance and unifying compression strategies for extreme model reduction.

[1] https://github.com/microsoft/BitBLAS

ETHICS STATEMENT

The research conducted in the paper conforms, in every respect, with the ICLR Code of Ethics.

REPRODUCIBILITY STATEMENT

We have provided implementation details in Section 4. We will also release all the code and models.

LLM USAGE STATEMENT

Large Language Models (LLMs) were used solely for polishing writing. They did not contribute to the research content or scientific findings of this work.

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
