# Progressive Binarization with Semi-Structured Pruning for LLMs

## Contents

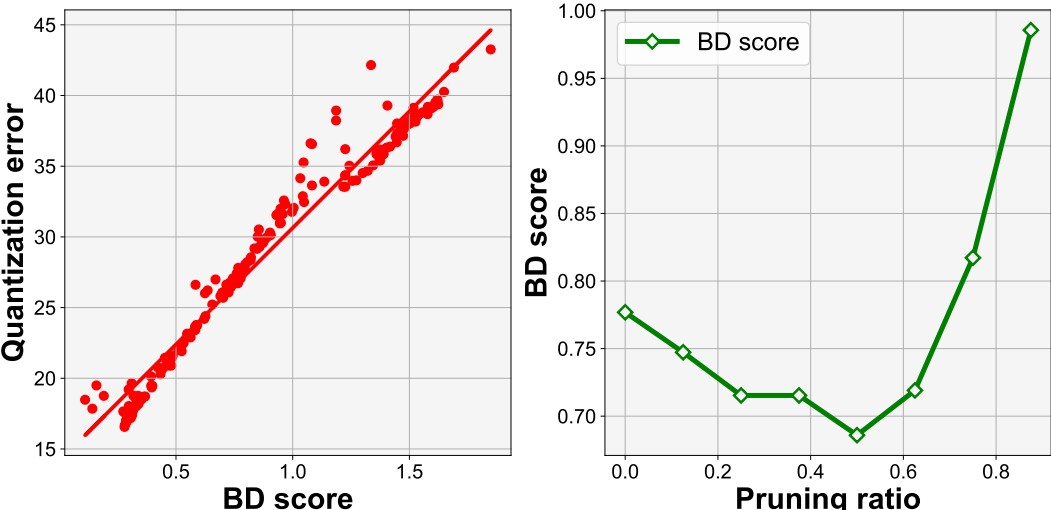

Figure 1: **Left**: The left side shows the relationship between quantization error and BD score, where they exhibit a positive correlation. **Right**: The right side shows the BD score of the same weight matrix at different pruning ratios, where the BD score only decreases at an appropriate pruning ratio.

## A  ANALYSIS OF BINARIZATION DIFFICULTY

After the standard binarization process, we define the quantization error $\mathcal{L}_1$ after binarization as:

$$\mathcal{L}_1 = \|\mathbf{W} - \widehat{\mathbf{W}}\|_F^2, \quad \text{where } \widehat{\mathbf{W}} = \alpha\mathbf{B} + \mu. \tag{1}$$

$\mathcal{L}_1$ represents the difference between the binarized matrix and the full-precision matrix. It can be observed that during the binarization process, the row-wise scaling factor $\alpha$ is used to approximate the magnitude of elements in each row, and the quantization error $\mathcal{L}_1$ varies depending on the weight distribution. Intuitively, when using the row-wise scaling factor approach for binarization, weight matrices with more dispersed row-wise distributions tend to face greater binarization difficulty, which in turn results in larger quantization errors.

**Binarization Difficulty Score.**  We revisit the binarization process and propose a new metric, the Binarization Difficulty (BD) score, which quantifies the inherent difficulty of binarizing a given weight matrix. This score not only provides a measure of how challenging the binarization task is, but also serves as an indicator of the potential quantization error that may arise as a result of the binarization process. The definition of BD score is as follows:

$$BD = \frac{1}{n}\sum_{i=1}^{n} \text{Var}\left(|\mathbf{W}_{i.} - \mu_i|\right), \tag{2}$$

where $\mathbf{W}$ represents the given weight matrix. We take into account the redistribution of the weight matrix caused by standard binarization in the calculation of the BD score, and apply absolute value processing before computing the variance. BD score quantifies the difficulty of binarization, where higher values of BD score indicate more dispersed weight distributions and thus greater binarization challenges.

To validate the effectiveness of the BD score, we performed binarization experiments on different weight matrices and analyzed the relationship between the BD score and quantization error. The results are shown in Figure 1 left. As observed, there is a positive correlation between the BD score and quantization error. We hope that the BD score can serve as a guiding metric in the process of binarization.

## B  IMPACT OF PRUNING ON BINARIZATION DIFFICULTY

**Pruning Enhances Binarization.**  We then attempt to apply the $N : M$ sparsity, a semi-structured pruning method, to prune the model. After pruning, we calculate the BD score of the remaining

weights in the weight matrix. Figure 1 right shows the BD score of a weight matrix at different pruning ratios. We observe that an appropriate pruning ratio can reduce the difficulty of binarization, leading to lower quantization errors in the binarization phase. Of course, pruning introduces additional pruning errors. In our experiments, we recorded the errors from both pruning error and binarization error, as well as their total error. It is evident that when we use an appropriate pruning ratio, the total error is lower than the binarization error without pruning. This is a crucial finding, suggesting that by combining pruning and binarization techniques, we can effectively reduce errors during model compression, thereby preserving both model performance and efficiency.

## C    ILLUSTRATIVE EXAMPLE: EFFECT OF PRUNING BEFORE BINARIZATION

To demonstrate the benefit of pruning before binarization, we construct a toy example using an $8 \times 8$ weight matrix $W$ and a $4 \times 8$ input matrix $X$, under a 6:8 pruning ratio. We compare the reconstruction errors of two strategies: direct binarization vs. pruning followed by binarization.

ORIGINAL WEIGHT MATRIX $W$

$$
\begin{bmatrix}
-1.5256 & -0.7502 & -0.6540 & -1.6095 & -0.1002 & -0.6092 & -0.9798 & -1.6091 \\
-0.7121 & 0.3037 & -0.7773 & -0.2515 & -0.2223 & 1.6871 & 0.2284 & 0.4676 \\
-0.6970 & -1.1608 & 0.6995 & 0.1991 & 0.8657 & 0.2444 & -0.6629 & 0.8073 \\
1.1017 & -0.1759 & -2.2456 & -1.4465 & 0.0612 & -0.6177 & -0.7981 & -0.1316 \\
1.8793 & -0.0721 & 0.1578 & -0.7735 & 0.1991 & 0.0457 & 0.1530 & -0.4757 \\
-0.1110 & 0.2927 & -0.1578 & -0.0288 & 2.3571 & -1.0373 & 1.5748 & -0.6298 \\
-0.9274 & 0.5451 & 0.0663 & -0.4370 & 0.7626 & 0.4415 & 1.1651 & 2.0154 \\
0.1374 & 0.9386 & -0.1860 & -0.6446 & 1.5392 & -0.8696 & -3.3312 & -0.7479
\end{bmatrix}
$$

INPUT MATRIX $X$

$$
\begin{bmatrix}
-0.0255 & -1.0233 & -0.5962 & -1.0055 & -0.2106 & -0.0075 & 1.6734 & 0.0103 \\
-0.7040 & -0.1853 & -0.9962 & -0.8313 & -0.4610 & -0.5601 & 0.3956 & -0.9823 \\
-0.5065 & 0.0998 & -0.6540 & 0.7317 & -1.4344 & -0.5008 & 0.1716 & -0.1600 \\
0.2546 & -0.5020 & -1.0412 & 0.7323 & -1.0483 & -0.4709 & 0.2911 & 1.9907
\end{bmatrix}
$$

6:8 PRUNING MASK

$$
\begin{bmatrix}
1 & 0 & 1 & 1 & 0 & 1 & 1 & 1 \\
1 & 0 & 1 & 0 & 1 & 1 & 1 & 1 \\
0 & 1 & 1 & 0 & 1 & 1 & 1 & 1 \\
1 & 0 & 1 & 1 & 0 & 1 & 1 & 1 \\
1 & 0 & 1 & 1 & 1 & 0 & 1 & 1 \\
0 & 1 & 1 & 0 & 1 & 1 & 1 & 1 \\
1 & 0 & 0 & 1 & 1 & 1 & 1 & 1 \\
0 & 1 & 0 & 1 & 1 & 1 & 1 & 1
\end{bmatrix}
$$

BINARIZED MATRIX WITHOUT PRUNING ($W_{\text{BIN-ONLY}}$)

$$
\begin{bmatrix}
-1.1053 & -1.1053 & -1.1053 & -1.1053 & -0.0998 & -1.1053 & -1.1053 & -1.1053 \\
-0.4908 & 0.6717 & -0.4908 & -0.4908 & -0.4908 & 0.6717 & 0.6717 & 0.6717 \\
-0.8402 & -0.8402 & 0.5632 & 0.5632 & 0.5632 & 0.5632 & -0.8402 & 0.5632 \\
0.2138 & 0.2138 & -1.2770 & -1.2770 & 0.2138 & -1.2770 & -1.2770 & 0.2138 \\
0.5973 & -0.3189 & 0.5973 & -0.3189 & 0.5973 & -0.3189 & 0.5973 & -0.3189 \\
-0.2787 & -0.2787 & -0.2787 & -0.2787 & 1.9660 & -0.2787 & 1.9660 & -0.2787 \\
-0.4327 & 0.9860 & -0.4327 & -0.4327 & 0.9860 & 0.9860 & 0.9860 & 0.9860 \\
0.6073 & 0.6073 & 0.6073 & -1.3983 & 0.6073 & -1.3983 & -1.3983 & -1.3983
\end{bmatrix}
$$

PRUNED AND BINARIZED MATRIX ($W_{\text{PRUNE+BIN}}$)

$$
\begin{bmatrix}
-1.5814 & 0.0000 & -0.7476 & -1.5814 & 0.0000 & -0.7476 & -0.7476 & -1.5814 \\
-0.5706 & 0.0000 & -0.5706 & 0.0000 & -0.5706 & 0.7944 & 0.7944 & 0.7944 \\
0.0000 & -0.9118 & 0.6542 & 0.0000 & 0.6542 & 0.6542 & -0.9118 & 0.6542 \\
0.4850 & 0.0000 & -1.2770 & -1.2770 & 0.0000 & -1.2770 & -1.2770 & 0.4850 \\
1.8793 & 0.0000 & -0.1479 & -0.1479 & -0.1479 & 0.0000 & -0.1479 & -0.1479 \\
0.0000 & -0.3831 & -0.3831 & 0.0000 & 1.9660 & -0.3831 & 1.9660 & -0.3831 \\
-0.6822 & 0.0000 & 0.0000 & -0.6822 & 1.0962 & 1.0962 & 1.0962 & 1.0962 \\
0.0000 & 1.2389 & 0.0000 & -1.3983 & 1.2389 & -1.3983 & -1.3983 & -1.3983
\end{bmatrix}
$$

RECONSTRUCTION ERROR COMPARISON

$$\|W - W_{\text{bin-only}}\|_F = 4.3851$$
$$\|W - W_{\text{prune+bin}}\|_F = 3.8359$$

These results confirm that applying N: M pruning before binarization helps eliminate less important weights, thereby enabling binarization to better preserve the structure of the remaining salient weights and reduce total approximation error.

## D  ERROR INCREASE FROM PRUNING ONE WEIGHT

Consider a loss function $E(\mathbf{w})$ minimized at $\mathbf{w}^*$. Using a second-order Taylor expansion around $\mathbf{w}^*$, the increase in error due to a small weight change $\delta\mathbf{w}$ is approximated by:

$$\Delta E \approx \frac{1}{2}\delta\mathbf{w}^\top H \delta\mathbf{w}, \tag{3}$$

where $H = \nabla^2 E(\mathbf{w}^*)$ is the Hessian matrix. The first-order term vanishes since $\nabla E(\mathbf{w}^*) = 0$ at a local minimum.

To prune the $q$-th weight, we enforce the constraint:

$$e_q^\top \delta\mathbf{w} + w_q = 0, \tag{4}$$

where $e_q$ is a unit vector with 1 at index $q$ and $w_q$ is the original value of the pruned weight.

We minimize $\Delta E$ subject to this constraint using the method of Lagrange multipliers. Define the Lagrangian:

$$\mathcal{L}(\delta\mathbf{w}, \lambda) = \frac{1}{2}\delta\mathbf{w}^\top H \delta\mathbf{w} + \lambda(e_q^\top \delta\mathbf{w} + w_q). \tag{5}$$

Setting the gradient of $\mathcal{L}$ with respect to $\delta\mathbf{w}$ to zero yields:

$$H\delta\mathbf{w} + \lambda e_q = 0 \quad \Rightarrow \quad \delta\mathbf{w}^* = -\lambda H^{-1} e_q. \tag{6}$$

Substituting into the constraint:

$$e_q^\top \delta\mathbf{w}^* + w_q = -\lambda e_q^\top H^{-1} e_q + w_q = 0, \tag{7}$$

Which gives:

$$\lambda = \frac{w_q}{(H^{-1})_{qq}}. \tag{8}$$

Thus, the optimal weight change is:

$$\delta \mathbf{w}^* = -\frac{w_q}{(H^{-1})_{qq}} H^{-1} e_q. \tag{9}$$

The corresponding increase in error is:

$$\Delta E = \frac{1}{2} \delta \mathbf{w}^{*\top} H \delta \mathbf{w}^* = \frac{w_q^2}{2(H^{-1})_{qq}}. \tag{10}$$

This quantity defines the saliency of weight $w_q$.

## E  DERIVATION OF THE OPTIMIZATION FORMULAS IN SPBO

Since simultaneously optimizing the pruning mask and binarization parameters is an NP-hard problem, we use a greedy algorithm to solve this issue. First, we derive the optimization formula for the binarization parameters without the pruning mask. The current definition of quantization error is:

$$\mathcal{L} = ||\mathbf{W}\mathbf{X} - \widehat{\mathbf{W}}\mathbf{X}||_F^2. \tag{11}$$

**Rewritten quantization error to decouple W and X**   We first rewrite the quantization error $\mathcal{L}$ to decouple $\mathbf{W}$ and $\mathbf{X}$, reducing the computational cost when calculating the quantization error. We define $\widetilde{\mathbf{W}}$ as $\widetilde{\mathbf{W}} = \mathbf{W} - \mu$. Then we rewrite the quantization error as:

$$\mathcal{L} = ||\mathbf{W}\mathbf{X} - \widehat{\mathbf{W}}\mathbf{X}||_F^2 \tag{12}$$

$$= ||\mathbf{X}(\widetilde{\mathbf{W}} - \alpha\mathbf{B})^\top||_F^2 \tag{13}$$

$$= \sum_i \sum_j (\sum_b \sum_k (\mathbf{X}_b)_{ik}(\widetilde{\mathbf{W}}_{jk} - \alpha_j\mathbf{B}_{jk}))^2. \tag{14}$$

The residual matrix is defined as $\mathbf{R} = \mathbf{W} - \mu - \alpha\mathbf{B}$ and further simplify $\mathcal{L}$:

$$\mathcal{L} = \sum_i \sum_j (\sum_b \sum_k (\mathbf{X}_b)_{ik}\mathbf{R}_{jk})^2 \tag{15}$$

$$= \sum_i \sum_j (\sum_b \sum_k \sum_l (\mathbf{X}_b)_{ik}(\mathbf{X}_b)_{il}\mathbf{R}_{jk}\mathbf{R}_{jl}) \tag{16}$$

$$= \sum_k \sum_l (\sum_b \sum_i (\mathbf{X}_b)_{ik}(\mathbf{X}_b)_{il})(\sum_j \mathbf{R}_{jk}\mathbf{R}_{jl}). \tag{17}$$

After that, we define the matrix $\mathbf{S}$ using the following formula:

$$\mathbf{S}_{kl} = \sum_b \sum_i (\mathbf{X}_b)_{ik}(\mathbf{X}_b)_{il}, \tag{18}$$

where $k = 1, 2, \ldots, m, l = 1, 2, \ldots, m$. Then we obtain the final simplified $\mathcal{L}$ as

$$\mathcal{L} = \langle \mathbf{S}, \mathbf{R}^\top\mathbf{R} \rangle_F = \mathrm{Tr}(\mathbf{R}\mathbf{S}\mathbf{R}^\top). \tag{19}$$

**Parameter Optimization Formula**   We use the quantization error $\mathcal{L}$ to update $\mu$:

$$\mathcal{L} = \sum_k \sum_l \mathbf{S}_{kl} \sum_j \mathbf{R}_{jk}\mathbf{R}_{jl} \tag{20}$$

$$= \sum_k \sum_l \mathbf{S}_{kl} \sum_j (\widetilde{\mathbf{W}}_{jk}\widetilde{\mathbf{W}}_{jl} \tag{21}$$

$$- \alpha_j(\mathbf{B}_{jk}\widetilde{\mathbf{W}}_{jl} + \mathbf{B}_{jl}\widetilde{\mathbf{W}}_{jk}) + \alpha_j^2\mathbf{B}_{jk}\mathbf{B}_{jl}) \tag{22}$$

$$= \sum_k \sum_l \mathbf{S}_{kl} \sum_j ((\mathbf{W}_{jk} - \mu_j)(\mathbf{W}_{jl} - \mu_j) \tag{23}$$

$$- \alpha_j(\mathbf{B}_{jk}(\mathbf{W}_{jl} - \mu_j) \tag{24}$$

$$+ \mathbf{B}_{jl}(\mathbf{W}_{jk} - \mu_j)) + \alpha_j^2\mathbf{B}_{jk}\mathbf{B}_{jl}). \tag{25}$$

To obtain the optimal solution for $\mu$, we take the partial derivative of $\mathcal{L}$ with respect to $\mu_j$, where $j = 1, 2, \ldots, n$:

$$\frac{\partial \mathcal{L}}{\partial \mu_j} = \sum_k \sum_l \mathbf{S}_{kl}(-\mathbf{W}_{jl} - \mathbf{W}_{jk} + 2\mu_j + \alpha_j \mathbf{B}_{jk} + \alpha_j \mathbf{B}_{jl}). \tag{26}$$

We set $\frac{\partial \mathcal{L}}{\partial \mu_j} = 0$ to get the optimal solution for $\mu_j$:

$$\mu_j = \frac{\sum_k \sum_l \mathbf{S}_{kl}(\mathbf{W}_{jk} - \alpha_j \mathbf{B}_{jk} + \mathbf{W}_{jl} - \alpha_j \mathbf{B}_{jl})}{2 \sum_k \sum_l \mathbf{S}_{kl}}, \tag{27}$$

$$\text{where } j = 1, 2, \ldots, n. \tag{28}$$

Then, we define the matrix $\mathbf{P}$ as:

$$\mathbf{P}_{kl} = \mathbf{W}_{jk} - \alpha_j \mathbf{B}_{jl}, \tag{29}$$

$$\text{where } k = 1, 2, \ldots, m, l = 1, 2, \ldots, m. \tag{30}$$

After that, we can simplify $\mu_j$ as

$$\mu_j = \frac{\sum_k \sum_l (\mathbf{S} \odot (\mathbf{P} + \mathbf{P}^\top))_{kl}}{2 \sum_k \sum_l \mathbf{S}_{kl}}, \tag{31}$$

$$\text{where } j = 1, 2, \ldots, n. \tag{32}$$

Since $\mathbf{S}$ is symmetric, we can further simplify the above equation as:

$$\mu_j = \frac{\sum_k \sum_l (\mathbf{S} \odot \mathbf{P})_{kl}}{\sum_k \sum_l \mathbf{S}_{kl}}, \quad \text{where } j = 1, 2, \ldots, n. \tag{33}$$

We can also express $\mu$ in a more compact vector form:

$$\mu = \frac{\mathbf{1}^\top \mathbf{S}(\mathbf{W} - \alpha \mathbf{B})^\top}{\mathbf{1}^\top \mathbf{S} \mathbf{1}}. \tag{34}$$

Similarly, we use the same quantization error to update $\alpha$:

$$\mathcal{L} = \sum_k \sum_l \mathbf{S}_{kl} \sum_j \mathbf{R}_{jk} \mathbf{R}_{jl} \tag{35}$$

$$= \sum_k \sum_l \mathbf{S}_{kl} \sum_j (\widetilde{\mathbf{W}}_{jk} \widetilde{\mathbf{W}}_{jl} \tag{36}$$

$$- \alpha_j(\mathbf{B}_{jk} \widetilde{\mathbf{W}}_{jl} + \mathbf{B}_{jl} \widetilde{\mathbf{W}}_{jk}) + \alpha_j^2 \mathbf{B}_{jk} \mathbf{B}_{jl}). \tag{37}$$

To obtain the optimal solution for $\alpha$, we take the partial derivative of $\mathcal{L}$ with respect to $\alpha_j$, where $j = 1, 2, \ldots, n$:

$$\frac{\partial \mathcal{L}}{\partial \alpha_j} = \sum_k \sum_l \mathbf{S}_{kl}(2\mathbf{B}_{jk} \mathbf{B}_{jl} \alpha_j - (\mathbf{B}_{jk} \widetilde{\mathbf{W}}_{jl} + \mathbf{B}_{jl} \widetilde{\mathbf{W}}_{jk})). \tag{38}$$

We set $\frac{\partial \mathcal{L}}{\partial \alpha_j} = 0$ to get the optimal solution for $\alpha_j$:

$$\alpha_j = \frac{\sum_k \sum_l \mathbf{S}_{kl}(\mathbf{B}_{jk} \widetilde{\mathbf{W}}_{jl} + \mathbf{B}_{jl} \widetilde{\mathbf{W}}_{jk})}{2 \sum_k \sum_l \mathbf{S}_{kl}(\mathbf{B}_{jk} \mathbf{B}_{jl})}, \tag{39}$$

$$\text{where } j = 1, 2, \ldots, n. \tag{40}$$

Then, we define the matrix $\mathbf{U}$ and $\mathbf{V}$ as:

$$\mathbf{U}_{kl} = \mathbf{B}_{jk} \widetilde{\mathbf{W}}_{jl}, \ \mathbf{V}_{kl} = \mathbf{B}_{jk} \mathbf{B}_{jl}, \tag{41}$$

where $k = 1, 2, \ldots, m$, $l = 1, 2, \ldots, m$. After that, we simplify $\alpha_j$ using $\mathbf{U}$ and $\mathbf{V}$:

$$\alpha_j = \frac{\sum_k \sum_l (\mathbf{S} \odot (\mathbf{U} + \mathbf{U}^\top))_{kl}}{2 \sum_k \sum_l (\mathbf{S} \odot \mathbf{V})_{kl}}. \tag{42}$$

Since $\mathbf{S}$ is symmetric, we can further simplify the above equation as

$$\alpha_j = \frac{\sum_k \sum_l (\mathbf{S} \odot \mathbf{U})_{kl}}{\sum_k \sum_l (\mathbf{S} \odot \mathbf{V})_{kl}}. \tag{43}$$

We can also express $\alpha$ in a more compact vector form:

$$\alpha = \frac{\mathrm{diag}(\mathbf{BS}(\mathbf{W} - \mu)^\top)}{\mathrm{diag}(\mathbf{BSB}^\top)}. \tag{44}$$

This is the parameter optimizing formula in the absence of a pruning mask. In the presence of a pruning mask, we directly prune the $\mathbf{W}$ and $\mathbf{B}$ matrices and then apply the optimizing formula. This is a simplified approach, where only the retained weights undergo binary approximation.

## F  ABLATION ON PRUNING VS. BINARIZATION

We conducted a set of controlled experiments to explicitly isolate and analyze the individual contributions of pruning and binarization. Specifically, we compare three configurations:

- **Pruning-only**: structured pruning with 4:8 pattern while keeping 16-bit full-precision weights.
- **Binarization-only**: 1-bit quantization without any pruning.
- **PBS²P (Ours)**: integration of 4:8 pruning and 1-bit binarization.

Table 1 reports the perplexity (lower is better) on the WikiText2 dataset across different LLaMA model sizes:

Table 1: Perplexity on WikiText2 across different LLaMA models for pruning-only, binarization-only, and our combined PBS²P method.

| Method | Sparsity | W-Bits | LLaMA-7B | LLaMA-13B | LLaMA-30B | LLaMA-65B |
|---|---|---|---|---|---|---|
| Dense (FP16) | – | 16-bit | 5.68 | 5.09 | 4.10 | 3.56 |
| Only Pruning | 4:8 | 16-bit | 8.61 | 7.40 | 6.17 | 5.38 |
| Only Binarization | – | 1-bit | 21.81 | 11.20 | 7.75 | 7.27 |
| PBS²P (Ours) | 4:8 | 1-bit | 10.78 | 9.24 | 7.19 | 6.39 |

From the above comparison, we make the following observations:

- **Pruning-only** methods retain full-precision (16-bit) weights and therefore maintain higher capacity, but offer limited compression benefit and are not suitable for ultra-low-bit deployment.
- **Binarization-only** methods drastically reduce model size, but suffer from substantial accuracy degradation, especially in larger models, due to the representational limitations of 1-bit weights.
- **PBS²P**, which jointly applies semi-structured pruning and 1-bit binarization, strikes a more favorable balance: it maintains strong perplexity while achieving extreme compression. For example, it improves over the binarization-only baseline by more than 50% on LLaMA-7B and LLaMA-13B.

These results clearly demonstrate that the performance gain of PBS²P arises not from either pruning or binarization alone, but from their synergistic integration. The progressive binarization pipeline and semi-structured pruning complement each other—enabling effective redundancy removal while preserving essential representational capacity.

## G EFFECTIVENESS OF COARSE- AND FINE-STAGE PRUNING STRATEGIES

This section provides supplemental experiments and extended analysis to further validate the roles of coarse-stage pruning ratio selection and fine-stage intra-layer pruning metrics discussed in the main paper.

**Coarse-stage Search Effectiveness.** Under the default 4:8 sparsity setting (i.e., 50% weight retention), the baseline performance is already competitive, leaving limited room for further improvement. As a result, the gains from coarse-stage search (CFS) may appear modest under light compression. However, as compression becomes more aggressive, the impact of CFS becomes significantly more pronounced.

To demonstrate this, we conduct additional experiments under a 3:8 sparsity setting (i.e., 37.5% weight retention). In this regime, CFS plays a crucial role in allocating sparsity across layers and selecting important weights. As shown in table 2, measured on the WikiText2 dataset (perplexity ↓), show that our full CFS pipeline substantially outperforms the ablated variant:

Table 2: Effect of CFS under 3:8 sparsity on WikiText2.

| Method | Sparsity | LLaMA-7B | LLaMA-2-7B | LLaMA-3-8B |
|---|---|---|---|---|
| PBS$^2$P w/o CFS | 3:8 | 25.61 | 25.03 | 45.60 |
| PBS$^2$P w/ CFS | 3:8 | 21.32 | 21.91 | 36.29 |

These results validate the importance of CFS under high compression ratios, where careful allocation of pruning budgets across layers becomes critical to maintaining model performance.

**Fine-stage Search Effectiveness.** For the intra-layer pruning metric, we adopt the Hessian-based criterion introduced in OBS. In Theorem 3.1 of our paper, we prove that this criterion is provably optimal under our pruning objective, minimizing the error introduced by weight removal. This is a principled choice grounded in theoretical analysis, not just a heuristic.

Empirically, we compare this metric against alternatives such as magnitude-based pruning and Wanda. Even under moderate sparsity levels, the Hessian-based criterion consistently outperforms the others. While the improvement may appear incremental in some cases, this is due to the high baseline performance of PBS$^2$P—yet in absolute terms, the Hessian-based metric still achieves the best results.

**Summary.** Although the individual gains of each component may appear modest under light compression, both the coarse-stage pruning ratio selection and fine-stage intra-layer pruning metric are essential to the overall robustness and scalability of PBS$^2$P. Their contributions become increasingly evident as compression pressure grows, supported by both empirical results and theoretical guarantees.

## H COMPRESSION TIME ANALYSIS

**Compression Time on LLaMA-7B.** As reported in Table 3 of the main paper, our method takes 111 minutes to compress LLaMA-7B, with only 2 minutes spent on the coarse-grained layer-wise pruning ratio search. The remaining time is attributed to the SPBO optimization stage, which performs fine-grained, iterative pruning and binarization refinement. We emphasize that this one-time cost is acceptable given the model scale and the significant inference efficiency benefits gained post-compression.

**Scalability to Larger Models.** To assess scalability, we applied the same PBS$^2$P pipeline to larger LLaMA models. The breakdown of compression time is presented in table 3. As shown, while total compression time increases with model size, the growth is approximately linear and remains within a practically acceptable range. Since the compression is performed only once per model checkpoint and brings long-term inference efficiency benefits, the time cost is justified even for larger models.

Table 3: Compression time (in minutes) across different model sizes.

| Model | Total Time (min) | Coarse Search (min) | SPBO Optimization (min) |
|---|---|---|---|
| LLaMA-7B | 111 | 2 | 109 |
| LLaMA-13B | 173 | 2 | 171 |
| LLaMA-30B | 393 | 3 | 390 |

## I  ON THE APPLICABILITY OF THE CLOSED-FORM SOLUTION TO SBPO

**Theoretical Basis.** Under ideal conditions, the SBPO optimization objective admits a closed-form solution for the scaling parameters $\alpha$ and $\mu$. Specifically, given a fixed binary mask, the optimal values of $\alpha$ and $\mu$ can be computed analytically by minimizing the reconstruction error defined by the quantization objective. This allows for efficient one-step updates without iterative approximation, providing a principled and computationally efficient baseline for sparsity-aware quantization.

**Masking-Induced Sparsity Constraints.** In practice, the applicability of the closed-form update is limited by the sparsity constraints introduced by the quantization pipeline. Following designs used in recent methods such as BiLLM, ARB-LLM, and STBLLM, the quantization process separately treats salient and non-salient weights. For non-salient weights, group-wise binarization is applied based on weight magnitude, while a semi-structured N: M pruning mask is also enforced. As a result, the binary matrix $\mathbf{B}$ is simultaneously constrained by:

- A group-wise binarization mask (enforcing intra-group locality).
- A semi-structured N: M pruning mask (enforcing inter-layer sparsity).

When these two masking mechanisms overlap, the effective support for some weight groups becomes extremely sparse. In such cases, the local optimization problem becomes underdetermined, and key terms in the closed-form expression (e.g., denominators involving norms or inner products) can become zero or ill-conditioned, resulting in numerical instability or undefined behavior.

**Optimization Strategy.** To ensure stability under high sparsity, the current implementation adopts a progressive sparsification strategy combined with alternating updates for $\mathbf{B}$ and scaling parameters. This approach avoids direct computation of potentially unstable closed-form terms and maintains robust performance across a wide range of sparsity settings.

**Conclusion.** The closed-form solution remains valid in theory and can be applied in low-sparsity or mask-free regimes. However, under compound masking constraints, progressive and alternating optimization offers a more stable and generalizable solution. Future work may further explore hybrid strategies that selectively apply closed-form updates in well-conditioned regions while maintaining robustness elsewhere.

## J  COMPARISON WITH PRUNING-ONLY BASELINES

To assess the performance of our method under standard sparsity settings, we compare PBS$^2$P with several representative pruning-only baselines that operate at 4:8 semi-structured sparsity and retain full-precision weights. These baselines include SparseGPT Frantar & Alistarh (2023), Wanda Sun et al. (2024), DST Yuxin Zhang & Ji (2023), and ALPS Meng et al. (2024). As shown in table 4, all methods are evaluated on the WikiText2 dataset using perplexity as the metric (lower is better).

**Evaluation Setting.** Pruning-only baselines retain 16-bit weight precision and thus benefit from higher representational capacity. In contrast, PBS$^2$P integrates 4:8 semi-structured pruning with 1-bit quantization, achieving substantially higher compression. This comparison highlights the trade-off between compression and performance.

**Observations.** As expected, pruning-only methods outperform quantized ones in terms of perplexity due to the retention of full-precision weights. However, PBS$^2$P achieves competitive performance while operating under an ultra-low-bit budget. This demonstrates the effectiveness of combining pruning with aggressive quantization for highly compressed inference, maintaining an acceptable perplexity gap relative to full-precision pruning.

Table 4: Perplexity on WikiText2 under 4:8 sparsity for pruning-only baselines (16-bit) and our method (1-bit).

| Method | Sparsity | W-bits | LLaMA-7B | LLaMA-13B | LLaMA-30B | LLaMA-65B |
|---|---|---|---|---|---|---|
| Dense | – | 16-bit | 5.68 | 5.09 | 4.10 | 3.56 |
| SparseGPT | 4:8 | 16-bit | 8.61 | 7.40 | 6.17 | 5.38 |
| SparseGPT + DST | 4:8 | 16-bit | 8.32 | 7.05 | 6.10 | 5.12 |
| Wanda | 4:8 | 16-bit | 8.57 | 7.40 | 5.97 | 5.30 |
| Wanda + DST | 4:8 | 16-bit | 8.45 | 7.25 | 5.91 | 5.26 |
| ALPS | 4:8 | 16-bit | 8.12 | 6.81 | N/A | N/A |
| PBS$^2$P (Ours) | 4:8 | 1-bit | 10.78 | 9.24 | 7.19 | 6.39 |

Table 5: Perplexity comparison of RTN, GPTQ, BiLLM, ARB-LLM, and PBS$^2$P on the LLaMA and OPT families. The evaluation results demonstrate the perplexity performance on the PTB dataset across various model sizes.

| Settings | | | LLaMA-1 | | | | LLaMA-2 | | LLaMA-3 | OPT | | |
|---|---|---|---|---|---|---|---|---|---|---|---|---|
| Method | #Block | W-Bits | 7B | 13B | 30B | 65B | 7B | 13B | 8B | 1.3B | 2.7B | 30B |
| FP16 | - | 16 | 41.15 | 28.09 | 23.51 | 25.06 | 37.91 | 50.93 | 11.18 | 20.29 | 17.97 | 14.03 |
| RTN | - | 3 | 3.2e2 | 64.52 | 80.45 | 81.56 | 1.6e3 | 2.2e2 | 1.8e3 | 8.9e3 | 9.0e3 | 1.0e3 |
| GPTQ | 128 | 3 | 84.88 | 26.40 | 20.21 | 19.54 | 4.8e3 | 40.33 | 18.83 | 17.54 | 15.15 | 11.28 |
| RTN | - | 2 | 1.2e5 | 8.4e4 | 3.2e4 | 2.1e4 | 2.4e4 | 5.1e4 | 6.3e5 | 8.0e3 | 5.9e3 | 1.0e5 |
| GPTQ | 128 | 2 | 1.4e3 | 2.2e2 | 69.46 | 47.70 | 5.5e3 | 4.1e2 | 717.23 | 1.1e2 | 58.38 | 14.18 |
| RTN | - | 1 | 1.5e5 | 1.9e6 | 1.4e4 | 6.8e4 | 9.9e4 | 3.8e4 | 7.6e5 | 1.1e4 | 2.8e4 | 5.4e3 |
| GPTQ | 128 | 1 | 1.2e5 | 1.0e5 | 1.0e4 | 2.0e4 | 6.6e4 | 2.7e4 | 9.7e5 | 6.5e3 | 8.4e3 | 7.1e3 |
| BiLLM | 128 | 1.11 | 3.7e2 | 84.87 | 43.10 | 44.68 | 5.2e3 | 3.0e2 | 87.25 | 1.1e2 | 88.52 | 21.41 |
| ARB-LLM | 128 | 1.11 | 1.9e2 | 54.38 | 34.65 | 32.20 | 389.59 | 1.9e2 | 45.49 | 43.34 | 31.77 | 16.88 |
| PBS$^2$P | 128 | 0.80 | **80.27** | **34.01** | **26.68** | **27.54** | **67.74** | **68.90** | **16.15** | **39.40** | **29.40** | **13.04** |
| PBS$^2$P | 128 | 0.70 | **87.69** | **38.11** | **28.38** | **29.12** | **75.22** | **79.59** | **18.65** | **46.98** | **31.71** | **17.14** |
| PBS$^2$P | 128 | 0.55 | **134.63** | **54.06** | **32.47** | **33.36** | **119.30** | **114.40** | **25.86** | **76.55** | **42.85** | **18.73** |

# K MORE EXPERIMENTAL RESULTS

We present more experimental results related to perplexity. In Table 5, we show the PTB test results of our method PBS$^2$P compared to other baseline methods. It is evident that our method outperforms the existing SOTA binarization methods. Table 6 displays the results on the C4 dataset, where our method continues to perform the best, and we also achieve a lower average bit count.

# L DIALOG EXAMPLES

In Figure 2, we present several dialogue results, including methods such as BiLLM, ARB-LLM, and PBSP. These methods are applied to the LLaMA-13B and LLaMA-2-13B models, and the corresponding dialogue results are shown.

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

Table 6: Perplexity comparison of RTN, GPTQ, BiLLM, ARB-LLM, and PBS$^2$P on the LLaMA and OPT families. The evaluation results demonstrate the perplexity performance on the C4 dataset across various model sizes.

| | Settings | | LLaMA-1 | | | | LLaMA-2 | | LLaMA-3 | OPT | | |
|---|---|---|---|---|---|---|---|---|---|---|---|---|
| Method | #Block | W-Bits | 7B | 13B | 30B | 65B | 7B | 13B | 8B | 1.3B | 2.7B | 30B |
| FP16 | - | 16 | 7.34 | 6.80 | 6.13 | 5.81 | 7.26 | 6.73 | 9.45 | 16.07 | 14.34 | 11.45 |
| RTN | - | 3 | 28.24 | 13.24 | 28.58 | 12.76 | 3.8e2 | 12.50 | 5.7e2 | 5.0e3 | 1.1e4 | 1.0e3 |
| GPTQ | 128 | 3 | N/A | 7.15 | 6.51 | 6.03 | 7.94 | 7.05 | 17.68 | 16.11 | 14.16 | 10.91 |
| RTN | - | 2 | 1.1e5 | 5.8e4 | 2.7e4 | 2.2e4 | 3.0e4 | 5.1e4 | 7.7e5 | 7.4e3 | 7.3e3 | 6.1e4 |
| GPTQ | 128 | 2 | 79.06 | 18.97 | 14.86 | 10.23 | 35.26 | 19.65 | 3.9e2 | 63.05 | 35.80 | 12.92 |
| RTN | - | 1 | 1.9e5 | 9.94 | 1.3e4 | 1.3e5 | 1.1e5 | 4.6e4 | 1.4e6 | 1.0e4 | 2.3e4 | 5.0e3 |
| GPTQ | 128 | 1 | 1.8e5 | 1.0e5 | 9.5e3 | 2.3e4 | 6.7e4 | 1.9e4 | 1.1e6 | 6.3e3 | 6.7e3 | 7.9e3 |
| BiLLM | 128 | 1.11 | 46.96 | 16.83 | 12.11 | 11.09 | 39.38 | 25.87 | 61.04 | 64.14 | 44.77 | 16.17 |
| ARB-LLM | 128 | 1.11 | 17.92 | 12.48 | 10.09 | 8.91 | 20.12 | 14.29 | 35.70 | 28.19 | 21.46 | 13.34 |
| PBS$^2$P | 128 | 0.80 | **9.52** | **8.11** | **7.16** | **6.63** | **9.43** | **8.44** | **15.62** | **25.62** | **20.78** | **12.69** |
| PBS$^2$P | 128 | 0.70 | **10.35** | **8.96** | **7.83** | **7.04** | **10.43** | **9.20** | **17.48** | **30.12** | **22.73** | **13.04** |
| PBS$^2$P | 128 | 0.55 | **13.06** | **11.44** | **9.33** | **8.52** | **13.12** | **11.19** | **22.90** | **46.07** | **30.33** | **14.21** |

LLaMA-13B

| | |
|---|---|
| **Prompt :** | Texas is a state of ___ |
| **BiLLM-1.1bit :** | Texas is a state of the United States. Texas is the largest state in the United States. |
| **ARB-LLM-1.1bit :** | Texas is a state that is famous for its oil and gas. |
| **PBS²P-0.8bit:** | Texas is a state of the United States. It is the second largest state in the United States. It is also the second most populated state in the United States. |
| **Prompt :** | Nile River is located in ___ |
| **BiLLM-1.1bit :** | Nile River is located in the heart of the city. |
| **ARB-LLM-1.1bit :** | Nile River is located in the Nile River. |
| **PBS²P-0.8bit:** | Nile River is located in the northern part of the African continent. It is the longest river in the continent. It is the only river in the continent that flows from one end to the other. |
| **Prompt :** | Having worked tirelessly on the project for days, her eyes were strained, her mind fatigued, and her hands sore from typing. Seeing her state, I recommended she ___ |
| **BiLLM-1.1bit :** | The project was completed, and the project was completed. |
| **ARB-LLM-1.1bit :** | Seeing her state, I recommended she 1) take a break, 2) take a nap, 3) take a shower, 4) take a walk, 5) take a break, 6) take a nap, 7) take a walk, 8) take a nap, |
| **PBS²P-0.8bit:** | Seeing her state, I recommended she 1) take a break, 2) get a massage, 3) take a walk, 4) get a manicure, 5) get a pedicure, 6) get a facial, 7) get a hair treatment, 8) get a makeover, 9) get a new wardrobe, 10) get a new car |

LLaMA-2-13B

| | |
|---|---|
| **Prompt :** | I fail on my math exam, and I am uncertain about what to do next. What should I do? |
| **BiLLM-1.1bit :** | I am uncertain about what to do next. |
| **ARB-LLM-1.1bit :** | I was taking math exam, and I have been taking the math exam since I was in junior high. |
| **PBS²P-0.8bit:** | You may not be able to get the grade you wanted on your math exam, but you may be able to get a grade you are happy with.The first thing you should do is take the time to figure out what you can do to improve your grade. |
| **Prompt :** | My friend is studying in Rome, and I want to visit. What's the easiest way to get there from Madrid? |
| **BiLLM-1.1bit :** | I am visiting in Paris for 10 days, and I'm thinking of travelling there from Madrid. |
| **ARB-LLM-1.1bit :** | I'm going to travel from Madrid to Paris. I'm thinking of going to Paris. |
| **PBS²P-0.8bit:** | It is a long way, and the best way to travel there from Madrid is by plane. It's 1200 km from Madrid to Paris, and it takes about 12 hours to get there by car. |
| **Prompt :** | After running a marathon, I was totally exhausted and out of breath. What's the best way to recover? |
| **BiLLM-1.1bit :** | A: 1. Rest for a week. 2. Do some exercise. 3. Have a good diet. 4. Have a good sleep. |
| **ARB-LLM-1.1bit :** | Most people would agree that the best way to recover after a marathon is to rest and relax. It is also good to have a rest and relaxation, because it is good to rest and relaxation. I am a big fan of marathon. |
| **PBS²P-0.8bit:** | There are many ways to recover from a marathon. Here are some of them. Rest: You need to take a break from running and give your body time to recover. Hydration: Drink plenty of water and electrolytes to replenish the fluids lost during the race. Massage: Get a massage to help with muscle soreness and tightness. Ice bath: Take an ice bath to reduce inflammation and help with muscle soreness. Heat pack: Use a heat pack to help with muscle soreness and tightness. Sleep: Get plenty of sleep to help. |

Figure 2: Conversation examples on LLaMA-13B (language supplementary) and LLaMA-2-13B(Q&A). We compare our best method PBS²P with BiLLM and ARB-LLM. Inappropiate and reasonable responses are shown in corresponding colors.