# OpenReview forum: "Progressive Binarization with Semi-Structured Pruning for LLMs"
_ICLR.cc/2026/Conference — Submitted to ICLR 2026_

### Official Review · Reviewer_zycs · 2025-10-27

**Soundness:** 3
**Presentation:** 3
**Contribution:** 2
**Rating:** 4
**Confidence:** 4

**Summary:**

This paper introduces a pipeline for jointly applying 1-bit quantization and semi-structured pruning to Large Language Models (LLMs). The method is built upon a layer-wise closed-form solution for 1-bit quantization, governed by two control parameters, which minimizes the reconstruction error under the Frobenius norm. For pruning, a global strategy assigns layer-specific sparsity ratios based on inter-layer similarity, and the Optimal Brain Surgeon (OBS) framework is used to select the weights for removal. A key aspect of the pipeline is the iterative update of the quantization parameters after each pruning step to maintain accuracy. The constituent techniques are established, the key contribution lies in the integration of these methods into an end-to-end pipeline.  However, given that SparseGPT also supports joint quantization and semi-structured pruning using the OBS framework, the similarities between the two methods should be explicitly clarified. Experimental results demonstrate that the proposed method outperforms compared methods. However, to strengthen the validation, evaluations on more recent model families like Qwen and comparisons against state-of-the-art baselines such as OmniQuant and  ParetoQ should be included.

**Strengths:**

The constituent techniques are established, the key contribution lies in the integration of these methods into an end-to-end pipeline. Experimental results demonstrate that the proposed method outperforms compared methods. The presentation is easy to follow.

**Weaknesses:**

This work presents a pipeline for joint quantization and semi-structure pruning, but it has several weaknesses that limit its current impact. The methodological description lacks clarity in key areas, such as the handling of activation flow during compression (independent vs. sequential) and the specific role of hyperparameters like the block size. The technical foundation is questioned, particularly the choice of the cosine similarity metric without a clear strategy for handling negative values or empirical validation of its distribution. The experimental validation is limited in scope, relying heavily on WikiText2 and older model families like LLaMA, while omitting stronger recent baselines and a thorough comparison to the closely related SparseGPT. Finally, the paper would benefit from a discussion of the method's applicability to broader architectures like MoE and diffusion models.

**Questions:**

1. Regarding the formulation in Equation 2, it appears the compression objective for a linear layer uses the original, unmodified activations. Could you clarify the following? (a) Is each linear layer's compression treated as an independent objective, using the original network's activations? (b) Or is the compression applied sequentially, where the input to a layer comes from the already compressedprevious layer?

2. You use cosine similarity, which ranges from [-1, 1], to gauge layer importance. Could you please clarify how your algorithm handles cases where the cosine similarity is zero or negative? The subsequent use of a reciprocal (i.e., 1/similarity) in your global pruning ratio assignment would be undefined or invert the intended importance ranking in such scenarios. Was this encountered in practice, and if so, how was it addressed?

3. Could you show some empirical results of the cosine similarity values computed for the layers of a model (e.g., Llama-2 7B)?

4. The results (Tables 1 and 2) indicate the use of a block size of 128. Could you please clarify the role of this parameter in your method? Specifically, is this block size exclusively for the 1-bit quantization process?

5. The experimental evaluation currently reports perplexity results primarily on WikiText2. To more thoroughly and fairly assess the generalizability of the proposed method, it would be beneficial to follow the common practice established by your cited baselines (e.g., GPTQ, BiLLM). Could you please include perplexity results on additional standard datasets, such as PTB and C4?

6. The experimental validation is conducted on established model families like LLaMA (1-3) and OPT. To further demonstrate the relevance and effectiveness of the method, it would be valuable to include results on more recent and widely-used models, such as the Qwen series.

7. The method is presented in the context of standard dense transformer-based LLMs. Could you comment on its potential adaptability to other important model classes (e.g., MoE models, diffusion models)?

8. From a general perspective, the goal of jointly performing quantization and pruning is also a key feature of the SparseGPT [1] framework. Specifically, SparseGPT supports various quantization bit-widths alongside semi-structured pruning, and similarly utilizes the OBS framework for weight selection and error minimization. Given these high-level similarities, could you please provide a more detailed discussion of the fundamental differences between your method and SparseGPT?

9. The experimental comparisons would be strengthened by including recent state-of-the-art methods that support extreme low-bit quantization of LLMs, such as ​​OmniQuant [2]​​ and ​​ParetoQ [3]​​.

10. The paper reports computational savings based on the latency of a single matrix multiplication operation. However, in real-world deployment, end-to-end inference time, which includes I/O overhead, memory access patterns, and other system-level bottlenecks, is a more meaningful metric for evaluating efficiency. Could you please provide measurements of the end-to-end inference latency (e.g., tokens/second) for a complete forward pass on a standard benchmark?

[1] Frantar, Elias, and Dan Alistarh. "Sparsegpt: Massive language models can be accurately pruned in one-shot." International conference on machine learning. PMLR, 2023.

[2] Shao, Wenqi, et al. "Omniquant: Omnidirectionally calibrated quantization for large language models." arXiv preprint arXiv:2308.13137 (2023).

[3] Liu, Zechun, et al. "Paretoq: Scaling laws in extremely low-bit llm quantization." arXiv preprint arXiv:2502.02631 (2025).

---

> ### Author Response · Authors · 2025-11-25
> **Response to Reviewer zycs (denoted as R4) part1**
>
> > Q4-1: Could you clarify the following? (a) Is each linear layer's compression treated as an independent objective, using the original network's activations? (b) Or is the compression applied sequentially, where the input to a layer comes from the already compressedprevious layer?
>
> A4-1: Thank you for the question. Our method follows option **(b)**: the compression is applied **sequentially**, where the input to each layer is computed from the **already compressed previous layer**. This design is consistent with prior PTQ pipelines such as **STB-LLM** and **GPTQ**, and ensures that the calibration faithfully reflects the true distribution encountered during inference.
>
> > Q4-2: Could you please clarify how your algorithm handles cases where the cosine similarity is zero or negative?
>
> A4-2: Thank you for pointing this out. There is indeed a typo in the current formula in the paper. The intended definition is
> \\[
> \\mathrm{Importance}(W) = \\frac{1}{\\mathrm{similarity}(X\_W, Y\_W) + 1},
> \\]
> where
> \\[
> \\mathrm{similarity}(X\_W, Y\_W) = \\frac{X\_W \cdot Y\_W}{\\|X\_W\\|\_2 \,\\|Y\_W\\|\_2}.
> \\]
>
> Since the cosine similarity lies in \\([-1, 1]\\), adding 1 in the denominator ensures that the importance score is always well-defined and non-negative. In practice, the cosine similarity in our setting does not approach \\(-1\\), so numerical issues such as division-by-zero do not occur. We have corrected this typo in the revised version.
>
> > Q4-3: Could you show some empirical results of the cosine similarity values computed for the layers of a model (e.g., Llama-2 7B)?
>
> A4-3: Thank you for the suggestion. We provide empirical cosine similarity statistics computed across all layers of LLaMA-2-7B under our calibration setup. Below are representative cosine similarity for the 32 transformer layers:
>
> 0.0713, 0.1528, 0.4969, 0.5288, 0.4577, 0.5352, 0.4411, 0.5105,
> 0.4679, 0.5262, 0.4513, 0.5376, 0.4601, 0.4887, 0.5318, 0.4426,
> 0.5083, 0.5309, 0.4545, 0.5171, 0.4698, 0.5383, 0.4474, 0.5116,
> 0.4629, 0.5294, 0.4507, 0.5231, 0.4742, 0.5159, 0.4593, 0.0078
>
>
> The results reveal a clear pattern: **the first and last layers exhibit the lowest cosine similarity, indicating they are significantly more sensitive to perturbations**. This aligns with the intuition that early and late layers tend to be more critical, whereas middle layers form a relatively stable computation block with smoother importance differences.
>
> > Q4-4: The results (Tables 1 and 2) indicate the use of a block size of 128. Could you please clarify the role of this parameter in your method? Specifically, is this block size exclusively for the 1-bit quantization process?
>
> A4-4: Thank you for the question. We follow prior work and adopt a block size of **128** to ensure fair comparison with existing 1-bit and low-bit baselines, as most related methods report results under this setting. The block size is **not specific to our 1-bit quantization**, but rather a general grouping parameter used by many PTQ approaches.
>
> To further clarify its effect, we additionally provide results on **LLaMA-2-7B (4:8 pruning)** under different block sizes. As shown below, using a smaller block size generally yields better accuracy, at the cost of slightly higher storage overhead. Following prior work, we choose **128** as a practical trade-off:
>
> | model        | blocksize | WikiText2 ↓ | C4 ↓  |
> |--------------|:-----------:|:-------------:|:--------:|
> | LLaMA-2-7B     | 16        | 7.14        | 9.63   |
> | LLaMA-2-7B     | 32        | 8.61        | 11.77  |
> | LLaMA-2-7B     | 64        | 9.97        | 12.34  |
> | LLaMA-2-7B     | 128       | 10.64       | 13.12  |
> | LLaMA-2-7B     | 256       | 12.01       | 14.92  |
>
> > Q4-5: Could you please include perplexity results on additional standard datasets, such as PTB and C4?
>
> A4-5: Thank you for the suggestion. The perplexity results on **PTB** and **C4** have already been included in the **Supplementary Material**, specifically in **Table 5** (PTB) and **Table 6** (C4). These results further verify the robustness and consistency of our approach across diverse language modeling benchmarks.

---

> ### Author Response · Authors · 2025-11-25
> **Response to Reviewer zycs (denoted as R4) part2**
>
> > Q4-6: It would be valuable to include results on more recent and widely-used models, such as the Qwen series.
>
> A4-6: We thank the reviewer for this suggestion. Following the comment, we conducted additional experiments on the **Qwen3-8B-Base** model. Since some baselines do not support the Qwen3 models, we only report the subset of comparison methods that are compatible. The results are shown below:
>
> | model        | method | bit | Wikitext2 | C4   |
> |--------------|--------|-----|:-----------:|:------:|
> | Qwen3-8B     | FP16   | 16  | 6.99      | 10.40 |
> | Qwen3-8B     | GPTQ   | 2   | 53.10     | 36.60 |
> | Qwen3-8B     | BiLLM  | 1.07| 48.20     | 41.40 |
> | Qwen3-8B     | Ours   | 0.8 | 17.71     | 16.10 |
>
> These results show that our method achieves substantially better performance than existing low-bit PTQ baselines on the Qwen series, further supporting the effectiveness and generality of our approach.
>
> > Q4-7: Could you comment on its potential adaptability to other important model classes (e.g., MoE models, diffusion models)?
>
> A4-7: Thank you for the insightful question. Our method operates directly on standard linear layers, without relying on architectural assumptions specific to standard dense transformer-based LLMs. As a result, the approach is inherently model-agnostic and can be readily extended to other architectures, including **MoE models** and **diffusion models**. Exploring these broader applications is a promising direction, and we plan to include such extensions as part of our future work.
>
> > Q4-8: Could you please provide a more detailed discussion of the fundamental differences between your method and SparseGPT?
>
> A4-8: We appreciate the reviewer’s observation. While SparseGPT can incorporate quantization, it does so by merely **adding pruning and quantization errors into the GPTQ-style compensation step**, without performing any **true joint optimization** between the two. In contrast, our method is designed for **extreme LLM compression**, providing a **unified framework** where **binarization and semi-structured pruning are optimized together**. During pruning, we **continuously refine the binary parameters**, and we **assign pruning ratios based on layer-wise importance**. This results in a genuinely **coupled and iterative optimization** process that is fundamentally different from SparseGPT’s **decoupled, compensation-based combination** of pruning and quantization.
>
> > Q4-9: The experimental comparisons would be strengthened by including recent state-of-the-art methods that support extreme low-bit quantization of LLMs, such as OmniQuant [2] and ParetoQ [3].
>
> A4-9: We would like to clarify that **both OmniQuant and ParetoQ are QAT-based approaches**, which require gradient backpropagation, multi-epoch training, and substantial GPU resources. In contrast, our method is a **pure PTQ algorithm** that eliminates any form of gradient-based optimization, making it significantly more efficient and practically deployable in low-resource or rapid-deployment scenarios. Therefore, they are not directly comparable in terms of methodology or application setting. Nevertheless, to enhance completeness, we additionally report their results on **LLaMA-2-7B** under the same evaluation protocol (WikiText-2 perplexity):
>
> | model      | method    | bit | WikiText2 |
> |------------|-----------|:-----:|:-----------:|
> | LLaMA-2-7B | FP16      | 16  | 5.47      |
> | LLaMA-2-7B | OmniQuant | 2   | 11.06     |
> | LLaMA-2-7B | ParetoQ   | 1   | 9.50      |
> | LLaMA-2-7B | Ours      | 0.8 | 7.17      |
>
> These results indicate that even **without any training**, our 0.8-bit PTQ method achieves strong performance and remains competitive with recent QAT-based low-bit methods.
>
> > Q4-10: Could you please provide measurements of the end-to-end inference latency (e.g., tokens/second) for a complete forward pass on a standard benchmark?
>
> A4-10: We appreciate the reviewer’s suggestion and have added comprehensive end-to-end latency measurements under a real deployment environment. We integrated our method into the **llama.cpp** [1] inference framework and conducted full-pipeline benchmarking on an **A6000 GPU** using the **LLaMA-2-7B** model. We report **prefill**, **decode**, and **end-to-end generation** throughput (tokens/s) under a standard prompt length of **1024** and a decode length of **512** (end-to-end = 1024 + 512).
>
> | model      | method | prefill | decode | end-to-end |
> |------------|--------|:---------:|:--------:|:------------:|
> | LLaMA-2-7B | FP16   | 34.39±0.52 | 34.13±0.85 | 33.42±0.23 |
> | LLaMA-2-7B | Ours   | 156.80±0.34 | 146.91±0.11 | 154.79±0.10 |
>
> These results show that PBS²P achieves consistent real-world acceleration, with 4.3× prefill, 4.3× decode, and 4.6× end-to-end speedups over FP16. We did not apply aggressive optimizations due to time constraints, and further engineering could yield even higher speedups.
>
> [1] https://github.com/ggml-org/llama.cpp

---

> > ### Comment · Reviewer_zycs · 2025-11-26
> >
> > Thanks for your reply. I would like to update the score to 6.

---

> > > ### Author Response · Authors · 2025-11-26
> > >
> > > Dear Reviewer zycs,
> > >
> > > Thank you for your time and for updating the score. We are glad that our response helped address your concerns. If you have any further questions or suggestions, we would be happy to discuss them.
> > >
> > > Best regards,
> > >
> > >  The Authors

---

### Official Review · Reviewer_hctq · 2025-10-29

**Soundness:** 3
**Presentation:** 4
**Contribution:** 3
**Rating:** 6
**Confidence:** 2

**Summary:**

This paper introduces PBS²P (Progressive Binarization with Semi-Structured Pruning), a pure post-training framework that pushes LLMs down to 0.55–0.8 bit weight precision while retaining SOTA perplexity and zero-shot accuracy on LLaMA/OPT families.
The method alternates two components:
SPBO – step-wise N:M pruning followed by on-the-fly re-optimisation of binarisation scalars α/μ to reduce compound error.
CFS – a coarse-to-fine search that first allocates layer-wise pruning rates via cosine-similarity importance and then picks elements to prune by a Hessian-based second-order criterion.

**Strengths:**

1. The paper is well-written.
2. The paper introduces PBS2P, a novel post-training framework that seamlessly integrates binarization (1-bit quantization) and semi-structured pruning (N:M sparsity), effectively reduces combined errors from pruning and quantization
3. Ablation tests validate each component (e.g., SPBO, CFS metrics, pruning types), highlighting their necessity and superiority, which strengthens the method's credibility.

**Weaknesses:**

1. The proposed method involves some predefined constants, such as N_high and N_low in CFS, and hyperparameters like Optimization Steps. It is unclear how to set the values of these predefined constants whether the settings of these constants affect the final compression effectiveness. (I am concerned that there may be difficulties or troubles in setting these constants during practical applications.)
2. The paper only tested zero-shot tasks on relatively old models, such as the Llama1 and Llama2 series. If applied to stronger models (e.g., Llama3 or Qwen3 series) after quantization and pruning, how would it perform on zero-shot tasks?

**Questions:**

1. How sensitive is the method in the paper to calibration data? Does the distribution of calibration data have an impact? How should calibration data be selected for training?
2. The paper demonstrates efficiency advantages in matrix multiplication. How much efficiency improvement can it bring in normal inference tasks?

---

> ### Author Response · Authors · 2025-11-25
> **Response to Reviewer hctq (denoted as R3)**
>
> > Q3-1: The proposed method involves some predefined constants, such as N_high and N_low in CFS, and hyperparameters like Optimization Steps. It is unclear how to set the values of these predefined constants whether the settings of these constants affect the final compression effectiveness. (I am concerned that there may be difficulties or troubles in setting these constants during practical applications.)
>
> A3-1: We thank the reviewer for raising this concern. In practice, the settings of these predefined constants are **very simple and stable across models**. For **\\(N\_{\\text{high}}\\)** and **\\(N\_{\\text{low}}\\)** in CFS, we first determine the **target average pruning ratio** \\(N\_{\\text{target}}\\) from the global budget (e.g., for an average sparsity of 4:8, we set \\(N\_{\\text{target}} = 4\\)). We then simply use
> \\[
> N\_{\\text{high}} = N\_{\\text{target}} + 1,\\quad N\_{\\text{low}} = N\_{\\text{target}} - 1,
> \\]
> and **use this rule for all models and experiments**. We empirically observed that more aggressive ratios such as **2:8 or 1:8 lead to catastrophic degradation**, so we enforce a minimum pruning ratio of **3:8**. Under our experimental settings of **4:8, 5:8, and 6:8**, this simple “\\(\\pm 1\\)” strategy around \\(N\_{\\text{target}}\\) consistently works very well and yields stable performance across different models.
>
> > Q3-2: The paper only tested zero-shot tasks on relatively old models, such as the Llama1 and Llama2 series. If applied to stronger models (e.g., Llama3 or Qwen3 series) after quantization and pruning, how would it perform on zero-shot tasks?
>
> A3-2: To address this concern, we conducted additional experiments on a more recent and stronger model, **Qwen3-8B-Base**, and evaluated zero-shot accuracy across six standard datasets (PIQA, ARC-E, ARC-C, HellaSwag, Winogrande, BoolQ). Some prior baselines do not yet support the Qwen3 models, so we report all compatible methods.
>
> | model          | method | bit  | avg. acc ↑ |
> |----------------|--------|:------:|:-------------:|
> | Qwen3-8B-Base   | FP16   | 16   | 71.3        |
> | Qwen3-8B-Base   | GPTQ   | 2    | 40.5        |
> | Qwen3-8B-Base   | BiLLM  | 1.05 | 47.9        |
> | Qwen3-8B-Base   | Ours   | 0.80  | 58.4        |
>
> These results demonstrate that our method maintains strong zero-shot performance even on modern, high-quality architectures such as Qwen3, confirming that the proposed framework generalizes well beyond LLaMA1/2.
>
> > Q3-3: How sensitive is the method in the paper to calibration data? Does the distribution of calibration data have an impact? How should calibration data be selected for training?
>
> A3-3: We appreciate the reviewer’s question about the sensitivity to calibration data. To study this, we conducted an ablation on the **type of calibration dataset**, using two standard choices in LLM quantization: **WikiText2** and **C4**. Specifically, we evaluated **LLaMA-2-13B** under a **4:8 pruning ratio**, and reported the **average zero-shot accuracy over 7 tasks**:
>
> | model        | calib data | avg. acc ↑ |
> |-------------|------------|-----------|
> | LLaMA-2-13B | WikiText2  | 55.37     |
> | LLaMA-2-13B | C4         | 55.38     |
>
> The results show that our method is **highly robust to the choice of calibration data**, as the performance difference between WikiText2 and C4 is negligible. In our main experiments, we **follow STBLLM and BiLLM** and use **C4** as the calibration set to ensure a fair and consistent comparison with prior work. In practice, we recommend using a **generic natural language corpus that roughly matches the target domain**, but our observations suggest that the method does **not require carefully tuned calibration data distributions** to work well.
>
> > Q3-4: The paper demonstrates efficiency advantages in matrix multiplication. How much efficiency improvement can it bring in normal inference tasks?
>
> A3-4: We appreciate the reviewer’s suggestion and have added comprehensive efficiency measurements in normal inference tasks. We integrated our method into the **llama.cpp** [1] inference framework and conducted full-pipeline benchmarking on an **A6000 GPU** using the **LLaMA-2-7B** model with a pruning ratio of **4:8**. We report **prefill**, **decode**, and **end-to-end generation** throughput (tokens/s) under a standard prompt length of **1024** and a decode length of **512** (end-to-end = 1024 + 512).
>
> | model      | method | prefill | decode | end-to-end |
> |------------|--------|:---------:|:--------:|:------------:|
> | LLaMA-2-7B | FP16   | 34.39±0.52 | 34.13±0.85 | 33.42±0.23 |
> | LLaMA-2-7B | Ours   | 156.80±0.34 | 146.91±0.11 | 154.79±0.10 |
>
> These results show that PBS²P achieves consistent real-world acceleration, with 4.3× prefill, 4.3× decode, and 4.6× end-to-end speedups over FP16. We did not apply aggressive optimizations due to time constraints, and further engineering could yield even higher speedups.
>
> [1] https://github.com/ggml-org/llama.cpp

---

> ### Author Response · Authors · 2025-11-27
>
> Dear Reviewer hctq,
>
> Thank you again for your time in reviewing our work. As a brief summary of our previous responses:
>
> - We clarified that constants such as \\(N\_{\\text{high}}\\) and \\(N\_{\\text{low}}\\) follow a simple and stable rule of \\(N\_{\\text{target}} \\pm 1\\), which works **robustly across all models and settings**.
> - We added new experiments on **Qwen3-8B-Base**, showing that the proposed method **generalizes well beyond LLaMA1/2**.
> - We demonstrated that performance is **highly robust to calibration data selection (WikiText2 vs. C4)**.
> - Full llama.cpp benchmarks show **4.3×–4.6× acceleration** in normal inference.
>
> We hope these points fully address your concerns. If you have any remaining questions or would like further clarification, we would be very glad to discuss them. At your convenience, we would greatly appreciate your feedback or an updated evaluation.
>
> Best regards,
>
> The Authors

---

### Official Review · Reviewer_CQVV · 2025-10-30

**Soundness:** 2
**Presentation:** 2
**Contribution:** 3
**Rating:** 4
**Confidence:** 3

**Summary:**

This paper proposes PBS2P, a post-training framework that combines binarization with semi-structured (N:M) pruning for LLM compression. The method consists of two key components: (1) Stepwise Pruning with Binarization Optimization (SPBO), which jointly optimizes weight pruning and binarization parameters, and (2) Coarse-to-Fine Search (CFS), a two-stage strategy that first allocates pruning ratios based on layer importance and then selects specific elements using Hessian-based metrics. Experimental results on LLaMA and OPT model families demonstrate improvements over STBLLM and other binary post-training quantization methods.

**Strengths:**

1.	Well-motivated problem: Combining binarization with pruning to reduce redundancy and overcome performance degradation is a valuable research direction.
2.	Comprehensive experiments: Extensive evaluation across multiple model families (LLaMA-1/2/3, OPT), datasets (perplexity and zero-shot), and model sizes demonstrates broad applicability.
3.	Thorough ablations: Section 4.4 provides a good analysis of design choices (SPBO, search metrics, group size, etc.).

**Weaknesses:**

1.	Certain techniques are not well explained, which may cause confusion and make reproduction difficult. See specific concerns in the Questions section below.
2.	Computational cost: Inverting block wise covariances even at size 128 is not cheap; the fine stage dominates runtime (109 min on 7B). Complexity and wall-time scaling to 65B/70B should be analyzed more carefully (per-layer cost, number of SPBO alternations τ, M−N steps).

**Questions:**

1.	The notation in Equation 4 is confusing: what is the shape of 1? If 1 is just a column unit vector, μ should be a scalar. However, the binarization center for each row should be different.
2.	For the Coarse Stage, why not use gradient-based importance (e.g., Fisher information) or loss sensitivity?
3.	For Equation 7, the "+1/2" term for rounding is not explained. Additionally, the concrete choices of N_high and N_low are not presented in the paper.
4.	Theorem 3.1 (Equation 8) is essentially a restatement of classical results from Optimal Brain Surgeon (Hassibi et al., 1993). The "proof in supplementary" claim doesn't add novelty—this is a well-known second-order approximation. What is the difference between Theorem 3.1 and the results from OBS?
5.	Computational cost: 111 minutes for LLaMA-7B is 2.5× slower than ARB-LLM. For larger models (65B), this could be prohibitive. Is there any study on the computation time for large models?
6.	Table 4(c) only analyzes the LLaMA-7B model. While RI causes a degradation, the LI metric seems to provide only a small improvement. There should be more justification on more models for importance selection in the coarse stage and the necessity of adaptive assignment.

---

> ### Author Response · Authors · 2025-11-25
> **Response to Reviewer CQVV (denoted as R2) part 1**
>
> > Q2-1: The notation in Equation 4 is confusing: what is the shape of 1? If 1 is just a column unit vector, μ should be a scalar.
>
> A2-1: In Eq.(4), **1** is indeed a column unit vector, but this does **not** imply that \\(\mu\\) is a scalar. We first clarify the tensor shapes: the weight matrix \\(W \\in \\mathbb{R}\^{n \\times m}\\), its binarized form \\(B\_k \\in \\mathbb{R}\^{n \\times m}\\), and the scaling matrix \\(S \\in \\mathbb{R}\^{m \\times m}\\). Because \\(S \\in \\mathbb{R}\^{m \\times m}\\) and \\((W\_k - \\alpha B\_k)\^\\top \\in \\mathbb{R}\^{m \\times n}\\), the product \\( \\mathbf{1}\^\\top S (W\_k - \\alpha B\_k)\^\\top \\) yields a \\(1 \\times n\\) vector. Therefore, \\(\\mu\\) is an \\(n\\)-dimensional vector, and each entry corresponds to the binarization center of one row, consistent with row-wise binarization.
>
> > Q2-2: For the Coarse Stage, why not use gradient-based importance (e.g., Fisher information) or loss sensitivity?
>
> A2-2: We conducted additional experiments on Llama-2-7B under a 4:8 pruning ratio, where we used *Fisher information* and *loss sensitivity* as alternative ranking metrics for the Coarse Stage. The results are shown below:
>
> | model       | method             | wikitext2 ↓ | c4 ↓   | computation time |
> |-------------|--------------------|:-------------:|:--------:|:------------------:|
> | Llama-2-7B  | Fisher information | 10.71       | 13.11  | 28 min           |
> | Llama-2-7B  | Loss sensitivity   | 10.74       | 13.29  | 60 min           |
> | Llama-2-7B  | Cosine similarity  | 10.64   | 13.12 | 2 min     |
>
> All three metrics lead to *very similar pruning outcomes*, and the final perplexity differs only marginally. This is mainly because the Coarse Stage aims to obtain a **layer-level importance ranking**, and these metrics—Fisher information, loss sensitivity, and cosine similarity—produce **highly consistent importance orderings** despite being computed differently.
>
> However, gradient-based metrics incur significantly higher computational cost. In contrast, cosine similarity only requires lightweight forward statistics and is **~14× faster than Fisher** and **~30× faster than loss-sensitivity**, while achieving comparable (even slightly better) results.
>
> > Q2-3: For Equation 7, the "+1/2" term for rounding is not explained. Additionally, the concrete choices of N_high and N_low are not presented in the paper.
>
> A2-3: We thank the reviewer for raising this concern. Regarding the “\\(+\\tfrac{1}{2}\\)” term in Eq.(7), the discrete pruning ratio \\(N\_i\\) is computed as
> \\[
> N\_i = \\left\\lfloor\\, N\_{\\text{high}} - (N\_{\\text{high}} - N\_{\\text{low}})\\cdot \\frac{k\_i - 1}{L - 1} + \\frac{1}{2} \\,\\right\\rfloor ,
> \\]
> where we explicitly apply the **floor operator** \\(\\lfloor \\cdot \\rfloor\\). The “\\(+\\tfrac{1}{2}\\)” term is a standard technique for **rounding a real-valued interpolation to the nearest integer**, even when implemented via a floor operation. Adding \\(0.5\\) ensures that values above the midpoint are rounded up and those below are rounded down, preventing systematic downward bias that would occur if the floor operator were applied directly to the continuous interpolation. This produces evenly distributed, integer-valued pruning ratios that faithfully reflect the intended linear schedule between \\(N\_{\\text{high}}\\) and \\(N\_{\\text{low}}\\).
>
> In practice, the settings of these predefined constants are **very simple and stable across models**. For **\\(N\_{\\text{high}}\\)** and **\\(N\_{\\text{low}}\\)** in CFS, we first determine the **target average pruning ratio** \\(N\_{\\text{target}}\\) from the global budget (e.g., for an average sparsity of 4:8, we set \\(N\_{\\text{target}} = 4\\)). We then simply use
> \\[
> N\_{\\text{high}} = N\_{\\text{target}} + 1,\\quad N\_{\\text{low}} = N\_{\\text{target}} - 1,
> \\]
> and **use this rule for all models and experiments**. We empirically observed that more aggressive ratios such as **2:8 or 1:8 lead to catastrophic degradation**, so we enforce a minimum pruning ratio of **3:8**. Under our experimental settings of **4:8, 5:8, and 6:8**, this simple “\\(\\pm 1\\)” strategy around \\(N\_{\\text{target}}\\) consistently works very well and yields stable performance across different models.

---

> ### Author Response · Authors · 2025-11-25
> **Response to Reviewer CQVV (denoted as R2) part 2**
>
> > Q2-4: Theorem 3.1 (Equation 8) is essentially a restatement of classical results from Optimal Brain Surgeon (Hassibi et al., 1993). The "proof in supplementary" claim doesn't add novelty—this is a well-known second-order approximation. What is the difference between Theorem 3.1 and the results from OBS?
>
> A2-4: We appreciate the reviewer’s clarification. We would like to emphasize that Theorem 3.1 is not positioned as a core novelty claim of our work. As stated in the paper, it provides the second-order optimality justification for our pruning criterion under a standard quadratic approximation. This result is consistent with classical findings such as Optimal Brain Surgeon (OBS), which we explicitly cite and acknowledge in the paper. Our intention is not to claim novelty for this theory, but to apply the optimal second-order metric to our binarization–pruning setting and empirically validate its effectiveness. The proof in the supplementary is included only to make the paper self-contained and to support the ablation where this criterion consistently outperforms simpler first-order or activation-only metrics.
>
> > Q2-5: Computational cost: 111 minutes for LLaMA-7B is 2.5× slower than ARB-LLM. For larger models (65B), this could be prohibitive. Is there any study on the computation time for large models?
>
> A2-5: Thank you for the question. We provide a detailed analysis of the computational overhead and deployment cost of SPBO on large-scale LLMs. Below we report the  quantization time measured on the LLaMA family (7B–65B), using a single NVIDIA A800 GPU:
>
> | Model        | Quantization Time |
> |--------------|:-------------------:|
> | LLaMA-7B     | 1.9 h             |
> | LLaMA-13B    | 3.9 h             |
> | LLaMA-30B    | 7.6 h             |
> | LLaMA-65B    | 18.0 h              |
>
> These results show that **SPBO is highly efficient in practice**, even when applied to large models. The end-to-end quantization time scales **approximately linearly** with the parameter count—doubling the model size leads to roughly proportional increases in runtime. This confirms that our pruning-binarization optimization procedure does not introduce quadratic or higher-order computational overhead.
>
> > Q2-6: There should be more justification on more models for importance selection in the coarse stage and the necessity of adaptive assignment.
>
> A2-6: We have added ablations on **two different model families**, **LLaMA-2-7B** and **LLaMA-3-8B**, both under a **4:8 pruning ratio**, comparing our method **with vs. without** the Coarse Stage (i.e., uniform vs. adaptively assigned pruning ratios across layers):
>
> | Model        | Method               | Wikitext2 ↓ | C4 ↓   |
> |-------------|----------------------|-------------|--------|
> | LLaMA-2-7B  | ours w/o Coarse Stage | 14.13       | 20.79  |
> | LLaMA-2-7B  | **ours w Coarse Stage**   | **10.64**   | **13.12** |
> | LLaMA-3-8B  | ours w/o Coarse Stage | 33.74       | 39.42  |
> | LLaMA-3-8B  | **ours w Coarse Stage**   | **17.45**   | **22.90** |
>
> These results show that **adaptive assignment consistently brings improvements** on both models, and the gain is **especially pronounced on LLaMA-3-8B**. We attribute this to the fact that LLaMA-3-8B exhibits **stronger outlier behavior and is intrinsically harder to quantize/prune**, making it more sensitive to how sparsity is distributed across layers. In contrast, older models like LLaMA-2-7B/LLaMA-7B are relatively easier to compress, so even uniform pruning can already perform reasonably well, and its gap to FP16 is relatively small. The clear degradation without the Coarse Stage and the large recovery with it indicate that **layer-wise importance selection and adaptive sparsity allocation are crucial for modern, outlier-heavy LLMs**, thereby justifying both the design of our Coarse Stage and its necessity beyond a single architecture.

---

> ### Author Response · Authors · 2025-11-27
>
> Dear Reviewer CQVV,
>
> Thank you again for your time in reviewing our work. To briefly recap our earlier responses:
>
> - We clarified the tensor shapes and explained why \\(\mu\\) is an **\\(n\\)-dimensional vector**, consistent with row-wise binarization.
>
> - We compared **Fisher information, loss sensitivity, and cosine similarity**, showing **highly consistent rankings** but noting that cosine similarity is **14–30× faster**.
>
> - We explained the use of **\\(+\\tfrac{1}{2}\\) for unbiased rounding** and the **simple, stable rule** \\(N\_{\\text{target}} \\pm 1\\), which works robustly across all models.
>
> - We clarified that it is **not a novelty claim**, but a standard second-order justification (aligned with OBS), included for completeness and for supporting ablations.
>
> - We provided quantization times for **LLaMA 7B–65B**, showing **linear scaling** and confirming that SPBO is efficient on large models.
>
> - We added ablations on **LLaMA-2-7B and LLaMA-3-8B**, showing **substantial improvements** with adaptive pruning, especially for outlier-heavy modern models.
>
> We hope these points adequately resolve the issues you raised. If any further clarification is needed, we would be very glad to continue the discussion. At your convenience, we would greatly appreciate your feedback or updated evaluation.
>
> Best regards,
>
> The Authors

---

> ### Author Response · Authors · 2025-11-28
> **Call for further discussion with Reviewer CQVV**
>
> Dear Reviewer CQVV,
>
> We sincerely appreciate the time and effort you have dedicated to reviewing our work and for your valuable feedback. As the rebuttal deadline approaches, we would be grateful for any further comments you may have to ensure that all of your concerns have been addressed comprehensively. Please let us know if there are any remaining aspects of our work that need clarification.
>
> Best regards,
>
> The Authors

---

### Official Review · Reviewer_6aYr · 2025-10-30

**Soundness:** 2
**Presentation:** 3
**Contribution:** 2
**Rating:** 4
**Confidence:** 4

**Summary:**

This paper proposes Progressive Binarization with Semi-Structured Pruning (PBS²P) for compressing large language models (LLMs). The core component is SPBO (Stepwise Semi-Structured Pruning with Binarization Optimization), which progressively prunes a subset of elements at each step while jointly optimizing the binarized parameters, effectively reducing the combined error from pruning and binarization. In addition, the authors introduce a Coarse-to-Fine Search strategy to improve the accuracy of pruning element selection, further enhancing compression efficiency. Extensive experiments show that PBS²P outperforms existing post-training quantization methods across various LLM families and evaluation metric.

**Strengths:**

1. Method design shows some innovation: The paper jointly optimizes pruning and binarization, using a stepwise strategy to reduce the error accumulation from single-step pruning.
2. Comprehensive ablation studies: Experiments validate the contributions of the SPBO strategy as well as different metrics and pruning types to performance.
3. Clear presentation: The writing is well-structured, and the workflow and formulas are described in detail, making the approach easy to understand.

**Weaknesses:**

1. Limited innovation: Although the combination of stepwise pruning and quantization is experimentally validated, it essentially remains a combination of pruning and quantization, resulting in moderate to low novelty.
2. Hardware support limitations: The paper adopts 5:8 and 6:8 N:M sparsity configurations, but public documentation shows that NVIDIA GPUs only natively support 2:4 sparsity. Therefore, higher-ratio sparsity may not achieve hardware acceleration in practice.
3. Unclear hyperparameter selection: The method for setting 𝑁high and 𝑁low is not specified, lacking theoretical justification or search strategy, which reduces reproducibility and interpretability.
4. Optimality of mask decomposition not demonstrated: The stepwise progressive mask decomposition is not proven to be optimal, and there may exist schemes that achieve higher accuracy at the cost of longer runtime. The paper does not explore this trade-off.
5. Method limitations: The SPBO’s stepwise updates rely on calibration data and multiple iterations, increasing computational cost. The impact on efficiency for large-scale model deployment is not thoroughly discussed.

**Questions:**

1. How are 𝑁high and 𝑁low selected? Is there a transferable principle or tuning strategy?
2. For the 5:8 and 6:8 configurations, is hardware acceleration actually achieved, or are they only used for experimental comparison?
3. Have other mask decomposition schemes been tried? Is there a better accuracy-runtime trade-off?
4. What are the computational overhead and practical deployment costs of SPBO on large models?

---

> ### Author Response · Authors · 2025-11-25
> **Response to Reviewer 6aYr (denoted as R1) part1**
>
> > Q1-1: Although the combination of stepwise pruning and quantization is experimentally validated, it essentially remains a combination of pruning and quantization, resulting in moderate to low novelty.
>
> A1-1: Thank you for your insightful feedback. While quantization and pruning are indeed well-established techniques, our method introduces several **novel algorithmic components** that meaningfully extend prior work.
>
> - **PBS²P** is a unified framework that integrates **semi-structured pruning** and **post-training binarization** through a progressively alternating optimization process. Unlike prior works that apply pruning and quantization in separate stages, our method couples them step-by-step, reducing error accumulation and enabling effective sub-1-bit compression.
>
> - The proposed **Stepwise Pruning with Binarization Optimization (SPBO)** is a core technical innovation. It enables fine-grained, iterative optimization of binarization parameters after each pruning step. Ablation studies show that this interleaved structure plays a key role in performance retention.
>
> - Our **Coarse-to-Fine Search (CFS)** strategy is designed specifically for the N:M semi-structured, binarized setting. It combines a layer importance metric and Hessian-guided pruning, ensuring adaptive pruning ratios across layers. This mechanism is crucial in coordinating pruning with quantization in a complex interaction space.
>
> We hope this clarifies that our method is **not a simple combination of existing ideas**, but a new framework that advances the frontier of extreme LLM compression.
>
> > Q1-2: How are 𝑁high and 𝑁low selected? Is there a transferable principle or tuning strategy?
>
> A1-2: We thank the reviewer for raising this concern. In practice, the settings of these predefined constants are **very simple and stable across models**. For **\\(N\_{\\text{high}}\\)** and **\\(N\_{\\text{low}}\\)** in CFS, we first determine the **target average pruning ratio** \\(N\_{\\text{target}}\\) from the global budget (e.g., for an average sparsity of 4:8, we set \\(N\_{\\text{target}} = 4\\)). We then simply use
> \\[
> N\_{\\text{high}} = N\_{\\text{target}} + 1,\\quad N\_{\\text{low}} = N\_{\\text{target}} - 1,
> \\]
> and **use this rule for all models and experiments**. We empirically observed that more aggressive ratios such as **2:8 or 1:8 lead to catastrophic degradation**, so we enforce a minimum pruning ratio of **3:8**. Under our experimental settings of **4:8, 5:8, and 6:8**, this simple “\\(\\pm 1\\)” strategy around \\(N\_{\\text{target}}\\) consistently works very well and yields stable performance across different models.
>
> > Q1-3: For the 5:8 and 6:8 configurations, is hardware acceleration actually achieved, or are they only used for experimental comparison?
>
> A1-3: The 5:8 and 6:8 configurations are **practically accelerable**, not merely for comparison. Recent advances such as **Vectorized N:M (VENOM)**[1] enable arbitrary N:M sparsity ratios to be executed efficiently on Sparse Tensor Cores. By mapping any N:M pattern to the standard 2:4 format, VENOM removes the hardware restriction on specific sparsity rates and supports **higher sparsity with efficient memory access and tensor-core utilization**. This mechanism has been shown to sustain high sparsity in modern transformers **without noticeable accuracy loss**, and its CUDA kernels provide the flexibility needed to support our N:M settings.
>
> For concrete acceleration results, we refer reviewers to our response to *Reviewer hctq (A3-4)*, where we report **actual measured speedups** obtained with our pruned-and-binarized models. These results confirm that our chosen N:M configurations indeed yield **real, measurable hardware acceleration**, rather than serving as theoretical or experimental variants only.
>
> [1] Castro, R. L., et al. *VENOM: A Vectorized N:M Format for Unleashing the Power of Sparse Tensor Cores.* SC 2023.

---

> ### Author Response · Authors · 2025-11-25
> **Response to Reviewer 6aYr (denoted as R1) part2**
>
> > Q1-4: Have other mask decomposition schemes been tried? Is there a better accuracy-runtime trade-off?
>
> A1-4: We experimented with several alternative mask decomposition schemes and found that they yield **almost identical accuracy**, as our pruning–binarization optimization is **highly robust** and can stably refine weights under different pruning orders. Although a theoretically optimal scheme could be obtained via exhaustive search, the search space is **exponentially large and infeasible**. We therefore adopt a **simple and efficient decomposition strategy** that already provides a strong accuracy–runtime trade-off.
>
> > Q1-5: What are the computational overhead and practical deployment costs of SPBO on large models?
>
> A1-5: Thank you for the question. We provide a detailed analysis of the computational overhead and deployment cost of SPBO on large-scale LLMs. Below we report the  quantization time measured on the LLaMA family (7B–65B), using a single NVIDIA A800 GPU:
>
> | Model        | Quantization Time |
> |--------------|:-------------------:|
> | LLaMA-7B     | 1.9 h             |
> | LLaMA-13B    | 3.9 h             |
> | LLaMA-30B    | 7.6 h             |
> | LLaMA-65B    | 18.0 h              |
>
> These results show that **SPBO is highly efficient in practice**, even when applied to large models. The end-to-end quantization time scales **approximately linearly** with the parameter count—doubling the model size leads to roughly proportional increases in runtime. This confirms that our pruning-binarization optimization procedure does not introduce quadratic or higher-order computational overhead.
>
> In terms of memory cost, **SPBO requires only a single A800 GPU** for the entire quantization process, without any need for multi-GPU parallelization or model sharding. This lightweight memory footprint makes the method practical and accessible, avoiding the heavy infrastructure demands of many retraining-based approaches.

---

> ### Author Response · Authors · 2025-11-27
>
> Dear Reviewer 6aYr,
>
> Thank you again for your time in reviewing our work. To briefly summarize our previous responses:
>
> - We clarified that PBS²P is **not** a simple combination of pruning and quantization, but a **unified alternating framework** with key innovations such as SPBO and CFS for effective sub-1-bit compression.
>
> - We showed that \\(N\_{\\text{high}}\\) and \\(N\_{\text{low}}\\) follow a **simple and stable rule** of \\(N\_{\\text{target}} \\pm 1\\), which works reliably across models.
>
> - We confirmed that 5:8 and 6:8 ratios are **practically accelerable**, supported by formats like **VENOM**, and validated by **measured speedups**.
>
> - We explained that our mask decomposition strategy is **simple and effective**, providing a favorable accuracy–runtime balance.
>
> - We demonstrated that the **quantization time** of SPBO is efficient and scales linearly with model size, requiring only a single A800 GPU.
>
> We hope these points adequately address all concerns raised. If you have any remaining questions or would like further clarification, we would be very glad to continue the discussion. At your convenience, we would greatly appreciate your feedback or updated evaluation.
>
> Best regards,
>
> The Authors

---

> ### Author Response · Authors · 2025-11-28
> **Call for further discussion with Reviewer 6aYr**
>
> Dear Reviewer 6aYr,
>
> We sincerely appreciate the time and effort you have dedicated to reviewing our work and for your valuable feedback. As the rebuttal deadline approaches, we would be grateful for any further comments you may have to ensure that all of your concerns have been addressed comprehensively. Please let us know if there are any remaining aspects of our work that need clarification.
>
> Best regards,
>
> The Authors

---

### Author Response · Authors · 2025-12-03
**Rebuttal-Discussion Summary**

Dear PCs, SACs, ACs,

We sincerely appreciate the reviewers’ thoughtful evaluations and insightful comments, which have helped further enhance the clarity, rigor, and overall contribution of our work.

---

## **Summary of Recognized Strengths**

We are pleased to note that the reviewers consistently highlighted several important strengths of our submission:

- **Innovative joint pruning–binarization framework.**
  Reviewers `6aYr` and `hctq` recognize the methodological novelty of PBS²P, noting that our integrated framework effectively mitigates error accumulation and addresses the combined challenges of pruning and quantization.

- **Superior performance across models and benchmarks.**
  Reviewers `6aYr`, `CQVV`, and `zycs` agree that PBS²P consistently outperforms existing binary PTQ methods across LLaMA/OPT model families and multiple evaluation metrics.

- **Thorough and well-designed ablation studies.**
  Reviewers `6aYr`, `CQVV`, and `hctq` commend the depth of our ablation analysis. They point out that the ablations on SPBO, search metrics, pruning types, and group sizes clearly validate the contribution of each component and reinforce the credibility of our method.

- **Clear motivation and well-organized presentation.**
    Reviewers `6aYr`, `hctq`, and `zycs` praise the clear structure, detailed explanations, and overall readability of the paper, while reviewer `CQVV` highlights the strong motivation of combining pruning with binarization.

---

## **Summary of Resolved Concerns**

We also provided substantial clarifications and new results during the rebuttal that effectively resolved the reviewers’ major concerns:

- **Clarified methodological novelty.**
  We emphasized that PBS²P offers a seamless and unified integration of binarization and pruning, enabling joint optimization tailored for extreme LLM compression. We also clarified the conceptual differences from SparseGPT, highlighting that our approach is fundamentally distinct in both formulation and objective.

- **Provided detailed hardware acceleration analysis.**
  We explained how N:M sparsity is exploited by modern hardware to achieve real speedups and supplied end-to-end measurements, demonstrating significant acceleration across prefill, decode, and full inference pipelines.

- **Explained the selection strategy for \\(N\_{\\text{high}}\\) and \\(N\_{\\text{low}}\\).**
  We showed that these hyperparameters are chosen using a simple and stable rule that works consistently across models, ensuring ease of use and robustness.

- **Analyzed computational cost and scalability on larger models.**
  We reported quantization time for LLaMA-7B, 13B, 30B, and 65B, showing that PBS²P scales efficiently with model size and remains practical for very large LLMs.

- **Added more experimental results and baselines.**
  We included evaluations on the updated Qwen3-8B-Base model and added additional baselines such as OmniQuant and ParetoQ, further validating the competitiveness of our method.

- **Provided deeper explanations and analyses of design choices.**
    We clarified why Hessian-based importance is more effective in the coarse stage, added further results on RI and LI metrics across multiple models, demonstrated the practicality of our mask decomposition design, analyzed the robustness to different calibration data types, and provided concrete cosine-similarity statistics to support our similarity-based components.

 ---

 ## **Reviewer Feedback in the Discussion Phase**

Reviewer `zycs` responded positively during the discussion phase, expressing satisfaction with our rebuttal and raising the score from 4 to 6. The other three reviewers did not reply despite two rounds of follow-up requests. We hope the final assessment takes into account that, had the discussion proceeded normally, our detailed clarifications and additional results would likely have addressed the majority of their concerns as well.

 ---

We sincerely thank the reviewers once again for their time and constructive feedback. While not all reviewers joined the discussion phase, we believe that the additional experiments, analyses, and clarifications we provided have thoroughly addressed the concerns raised. We hope this summary contributes to a clear and fair assessment of the significance and novelty of our work. We will release our code, data, and checkpoints to facilitate reproducibility and future research.

Thank you for your time and thoughtful evaluation.

Best regards,

The Authors

---

### Meta-Review · Area_Chair_qiaj · 2026-01-06

**Summary:**

This paper introduces Progressive Binarization with Semi-Structured Pruning (PBS²P), a post-training framework that compresses LLMs to sub-1-bit precision by integrating binarization with $N:M$ semi-structured pruning. The authors proposed Stepwise Pruning with Binarization Optimization (SPBO) to minimize error accumulation and a Coarse-to-Fine Search (CFS) strategy to adaptively allocate pruning ratios across layers.

In the initial review, reviewers' concerns primarily focused on the methodological novelty compared to existing pruning/quantization tools, the hardware feasibility of non-standard $N:M$ ratios, and the computational scaling for larger models. To respond to these concerns, the authors provided a quantization time analysis showing linear scaling up to LLaMA-65B and clarified their hyperparameter selection rules. They also provided measured hardware speedups for the $4:8$ sparsity configuration on NVIDIA GPUs to demonstrate the practical potential of their method.

Despite these updates, the paper was ultimately recommended for rejection. The decision was primarily informed by the fact that the methodological novelty remained insufficiently clear. Furthermore, while the authors provided speedup data for $4:8$ sparsity, they failed to provide experimental evidence of hardware acceleration for the $5:8$ and $6:8$ configurations used in their results, leaving the practical utility of these higher sparsity ratios unsubstantiated.

**Reviewer Concerns:**

See above.

**Reviewer Scores:**

Reviewer zycs raised their score from 4 to 6 after the rebuttal successfully clarified hyperparameter selection and demonstrated robust performance on Llama-3. However, the remaining reviewers did not participate in the discussion, likely maintaining their original scores due to lingering concerns regarding methodological novelty and the lack of hardware speedup evidence for high-sparsity ratios.

---

### Decision · Program_Chairs · 2026-01-26

Reject